# Causal Effect Estimation with Learned Instrument Representations

## Abstract

In many applications, we aim to assess the impact of a policy or intervention on outcomes of interest using retrospective data. This setting is challenging due to unobserved confounding, which can bias causal estimates. One approach to address this issue—in statistics, econometrics, and epidemiology—is to use *instrumental variables* (IVs) within two-stage regression frameworks. An IV is a variable that influences the treatment but has no direct effect on the outcome and is independent of unobserved confounders. However, across many applications, suitable and valid IVs are difficult to find or may not be available at all. We propose a method for decomposing the observed variables to find a representation which satisfies the standard IV assumptions of relevance, exclusion restriction, and unconfoundedness. To implement this decomposition, we introduce a deep learning model, *ZNet*, with an architecture that mirrors the *structural causal model* of IVs and is compatible with a wide range of two-stage IV estimators. Our experiments demonstrate that ZNet can (i) recover ground-truth instruments when they exist and (ii) construct proxy latent instruments that reduce bias due to unobserved confounding when no explicit instruments are available. These results suggest that ZNet can be used as a plug-in module for causal effect estimation in general observational settings, regardless of whether the (untestable) assumption of unconfoundedness is satisfied.

## 1 Introduction

Randomized controlled trials (RCTs) are the gold standard for identifying the *causal* impact of an intervention or treatment policy in medicine and beyond (Bothwell et al., 2016). RCTs support causal conclusions since randomization ensures that treatment assignments are not influenced by variables which also affect the outcome of interest. However, we often want, or need, to evaluate treatment effects outside of an RCT. For instance, a clinical trial for some medical interventions can be unethical, and randomization of social interventions can be infeasible. Thus, there is also growing need to develop alternative evidence generation methods for settings where RCTs are prohibitively expensive and time-consuming.

In settings where an RCT is infeasible, it is common to utilize retrospective data along with causal inference methods that adjust for confounding in real-world treatment assignments. However, in many real-world datasets, the confounding factors are unobserved and testing for their existence is impossible (Imbens & Rubin, 2015). For instance, consider the problem of determining the effects of a newly available consumer AI tool for mobile heart monitoring on cardiovascular health using electronic health records (EHRs). While EHRs contain numerous variables about an individual's lab results, medications, and diagnoses, they may not account for lifestyle factors which influence both consumer choice to use the product and health outcomes. In this setting, traditional causal inference methods to determine the efficacy of the tool will produce biased estimates potentially obscuring the true health impact.

A common approach to account for unobserved confounding used across statistics, econometrics, and epidemiology is to use an *instrumental variable* (IV) that does not directly influence the outcome, but directly affects the treatment, which enables unbiased estimation of causal effects under certain conditions (Imbens & Rubin, 2015). Classic examples of IVs used in the literature include: in economics, geographical proximity to a college as an instrument for educational attainment in estimating the returns of schooling (Card, 1993) and draft lotteries as instruments to study the effect

of military service on long-term economic outcomes (Angrist, 1990). Genetic factors have been used as IVs in medicine since gene variants are often highly correlated with risk factors but do not directly influence outcomes associated with these risk factors (Davey Smith & Ebrahim, 2003; Davey Smith & Hemani, 2014). While such IVs can enable causal identification, they were all present in the initial dataset construction and known to the analyst. Moreover, candidate instruments may not be strong or valid. For example, the use of genetic IVs is challenged by the strength of their correlations with risk factors (Davies et al., 2015; Burgess et al., 2016), the availability of genetic data in large and representative populations, and the risk that pleiotropy induces bias in effect estimates (i.e., gene variants are associated with multiple traits which leaves pathways other than through the considered risk factor open). Together, this means that appropriate IVs are often hard to find in practice.

In this paper, we consider the construction of IVs automatically from the observed data. Several existing works select or refine candidate instruments, especially genetic factors, to improve downstream effect estimation (Kuang et al., 2020; Silva & Shimizu, 2017; Kang et al., 2016; Zhang et al., 2021; Burgess et al., 2016; Davies et al., 2015). These works do not remove the need for domain expertise as a candidate IV must be included in the observed data. There is a small body of existing works which learn variational distributions to construct IV representations from probabilistic associations (Yuan et al., 2022; Li & Yao, 2024; Cheng et al., 2023; Chou et al., 2024). We combine these efforts by learning a discriminatory decomposition of the feature space into confounder and instrumental components through a model architecture that encodes the structural causal model (SCM) of IVs with ZNet. If there are existing instruments, ZNet can recover representations highly correlated with these variables. In the absence of existing instruments, ZNet learns a representation that serves as an instrument. This automated instrument construction can mitigate the need to rely on domain expertise to circumvent untestable assumptions about unobserved confounders. We demonstrate the ability of our method to learn suitable and superior instrument representations for causal inference.

## 2 PRELIMINARIES

Let $Y \in \mathcal{Y} \subset \mathbb{R}$ denote a continuous outcome, $T \in \mathcal{T} \subset \mathbb{R}$ a treatment variable (discrete or continuous), and $C \in \mathcal{X} \subset \mathbb{R}^d$ a set of observed covariates associated with each unit. The treatment $T$ has a causal effect on the outcome $Y$, while the covariates $C$ may influence both $T$ and $Y$ (i.e., $C$ confound the relation of primary interest between $T$ and $Y$). In addition to these measured confounders $C$, there are unknown or *unobserved confounders* $U$, which induce spurious associations by simultaneously affecting both the treatment and outcome (Fig. 1a.). We assume that the outcome variable $Y$ is determined by the following SCM:

$$Y = \varphi(C, T) + e_Y(U), \ \ T = \psi(C) + e_T(U), \tag{1}$$

for some unknown functions $\varphi : \mathcal{X} \times \mathcal{T} \to \mathcal{Y}$ and $\psi : \mathcal{X} \to \mathcal{T}$. Following (Hartford et al., 2017), we assume that the unobserved confounders $U$ influence the outcome $Y$ and the treatment $T$ additively, via the "error" functions of U, $e_Y$ and $e_T$, respectively, which we henceforth denote $e_Y$ and $e_T$. As a result, the observational and interventional distributions generally differ, i.e., $\mathbb{E}[Y|C, T] = \varphi(C, T) + \mathbb{E}[e_Y|C, T] \neq \varphi(C, T) + \mathbb{E}[e_Y|C] = \mathbb{E}[Y|do(T), C]$. Thus, estimating the potential outcome associated with $T = t$ would lead to a *confounding bias* $\Delta(c, t)$, where

$$\Delta(c, t) = \mathbb{E}[Y|C = c, T = t] - \mathbb{E}[Y|C = c, do(T) = t], \forall c, t. \tag{2}$$

**De-confounding with instrumental variables (IVs).** A common method for removing the confounding bias $\Delta(c, t)$ is to use IV regression. In the classical IV setting, we assume access to an additional variable $Z$ that is not influenced by the unobserved confounders $U$, affects the treatment $T$, and has no direct effect on the outcome $Y$ (Verbeek, 2004; Angrist et al., 1996). Formally, given a set of observed confounders $C$, $Z$ is a valid IV if it satisfies the following conditions [1] :

> *Unconfoundedness:* $Z \perp e_Y | C$,
>
> *Exclusion restriction:* $Z$ only enters $\varphi$ through $T$
>
> *Relevance:* $Z \not\perp T | C$.

---

[1]The definition we provide here is that of a conditional instrument in order to match DeepIV (Hartford et al., 2017) the most general downstream IV estimator. Ultimately, we construct $Z, C$ to be independent in addition to these assumptions so that $Z$ is a IV without conditioning on $C$ as well.

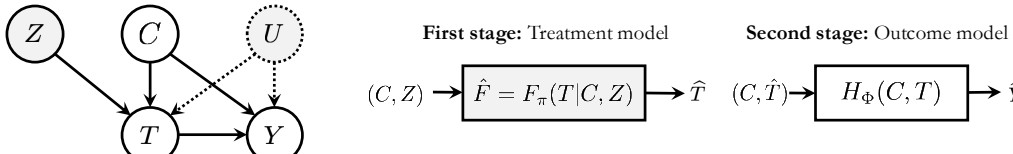

**(a)** Causal graph in the presence of IVs     **(b)** Two-stage framework for counterfactual prediction with IVs

Figure 1: **Illustration of the IV setting.** (a) Causal graph: nodes $Z, C, T, U$ and $Y$ represent the IV, covariates, treatment, unobserved confounders and outcome, respectively; (b) Two-stage framework for counterfactual prediction via IVs: $F_\pi$ and $H_\Phi$ are the learned treatment and outcome functions.

Under these conditions, the variable $Z$ can be used in a two-stage regression framework to estimate the effect of $T$ on $Y$. Under the additive model in (1), (Hartford et al., 2017) uses the instrument $Z$ to set up an inverse problem by relating the counterfactual $\mathbb{E}[Y|do(T), C]$ to observable distributions:

$$\mathbb{E}[Y|C, Z] = \mathbb{E}[\varphi(C, T) + e_Y|C, Z] = \mathbb{E}[\varphi(C, T)|C, Z] + \mathbb{E}[e_Y|C]$$
$$= \int \mathbb{E}[Y|do(T), C]\, dF(T|C, Z). \tag{3}$$

Thus, with $Z$, we can estimate the counterfactual $\mathbb{E}[Y|do(T), C]$ by learning models for the two observable functions $\mathbb{E}[Y|C, Z]$ and $F(T|C, Z)$. While this inverse problem is *ill-posed*, it provides a practical framework for estimating $\mathbb{E}[Y|do(T), C]$, and identification is possible under certain conditions (Newey & Powell, 2003)[2]. A typical two-stage regression first fits a model $\widehat{F}(T|C, Z)$, and then estimates $\mathbb{E}[Y|do(T), C]$ by replacing $F(T|C, Z)$ with $\widehat{F}(T|C, Z)$ in (3) (Fig. 1(b)).

Equation 3 is notably more general than traditional IV regression with linear models. Here we allow $\mathbb{E}[e_Y|C] \neq 0$, i.e. observed confounders can be correlated with unobserved errors. In a two stage least square regression (TSLS), no endogenous variables can remain for unbiased regression estimates (Verbeek, 2004). With either framework, we obtain conditional average treatment effects (CATE) and and average treatment effects (ATE):

$$\text{CATE}(C) = \mathbb{E}[Y|do(T) = 1, C] - \mathbb{E}[Y|do(T) = 0, C]$$
$$\text{ATE} = \mathbb{E}[Y|do(T) = 1] - \mathbb{E}[Y|do(T) = 0]$$

## 3   CONSTRUCTING INSTRUMENTS FROM DATA

To apply the standard two-stage IV framework described in Section 2, we typically have access to a valid instrument $Z$ among a collection of observed covariates $X$. In this case, we would set $C = X \setminus Z$. In our set up, instead of $Z$ being some known subset of the observed variables $X$, $Z$ is *learned* from data. We do not assume that instruments exist as a subset of the observed data. As illustrated in Figure 2, we derive from the observed variables $X$ two new sets, a confounder $C$ and an instrument $Z$, such that $Z$ satisfies the three key assumptions listed in Section

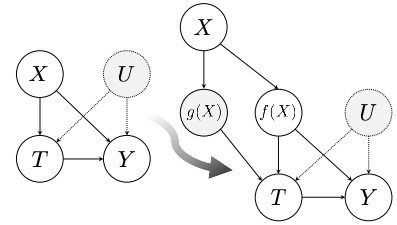

Figure 2: **Graph for constructed IVs**.

2. In the process, we learn a new SCM by learning new structural equations. We construct two functions (i.e., neural networks) $f, g$ that learn variable sets $C = f(X), Z = g(X)$ from $X$ such that $Y = \varphi'(C, T) + e_Y$ and $T = \psi'(C, Z) + e_T$ with fixed $T, Y$. This defines a new SCM where the instrument $Z$ is derived from the observed data $X$ without *a priori* being known or interpretable and the relationship between $T$ and $Y$ is unchanged.

Notice that this construction is suitable in general. First, suppose there is a subset of the variables that can serve as an instrument, $Z \subset X$. Then if we learn $g(X) = Z$ and $f(X) = X \setminus Z$, we succeed. Second, suppose there is a latent instrument learnable from the observed data. For example, imagine

---

[2]For example, in the linear case, Two-Stage Least Squares Regression (TSLS) allows for identification of causal effects.

the case of an emergency department. Randomly assigned providers have varying propensities to give treatments from their interpretation of the data observed for any patient but assignment does not affect outcomes directly. Such providers are assigned and take actions in ways that show up in the retrospective data: patient visit time, labs and tests run, symptoms listed in notes, etc. The provider influences $X$ and a representation of this influence might be inferred by $g$ to serve as $Z$. However, nothing about the construction of our model requires an *a posteriori* interpretation of $Z$. ZNet constructs an instrument representations even in the absence of true instruments. Sufficient satisfaction of the three IV conditions allows for IV regression independent of interpretation, and our derived instruments can be abstract exogenous representations of the feature space. This means our method can be used without domain knowledge of instrument existence and leverages existing instrumental variable relationships, should they exist.

The key idea of our proposed method is to force the desired causal dependencies through learned structural equations, i.e. we learn functions $f, g$ forcing IV conditions to hold with $Z = g(X)$ and $C = f(X)$. Relevance requires that we learn a variable $Z$ predictive of $T$. We therefore force non-zero covariance between $g(X)$ and $T$. Exclusion restriction requires that all direct influence of $X$ on $Y$ be captured by $C$. This is encouraged by forcing non-zero covariance between $f(X)$ and $Y$ and zero covariance between $f(X)$ and $g(X)$. The derived IV $g(X)$ will automatically be unconfounded and independent of the error $e_Y$ if derived from $X$ which is unconfounded by $U$. The assumption that observed variables are not influenced by $U$ is standard to allow for classical IV regression and straightforward IV generation (Yuan et al., 2022; Li & Yao, 2024; Cheng et al., 2023; Chou et al., 2024).

To allow for our method to produce an instrument even more generally when $X$ may be influenced by $U$, we add an additional constraint inspired by the following observation.

**Lemma 1.** *If $Z \sim \mathcal{N}(0, \sigma^2)$ and $Cov(Z, e_Y - \mathbb{E}[e_Y | X, T]) = 0$, then $Cov(Z, e_Y) = 0$.*

*Proof.* Notice that as $\mathbb{E}[Z] = 0$, we have

$$
\begin{aligned}
0 &= \mathrm{Cov}(Z, e_Y - \mathbb{E}[e_Y | X, T]) \\
&= \mathbb{E}[Z \cdot (e_Y - \mathbb{E}[e_Y | X, T])] - \mathbb{E}[Z] \cdot \mathbb{E}[(e_Y - \mathbb{E}[e_Y | X, T])] \\
&= E[Z \cdot (e_Y - \mathbb{E}[e_Y | X, T])] \\
&= \mathbb{E}[Z \cdot e_Y] - \mathbb{E}[Z] \cdot \mathbb{E}[e_Y | X, T] \\
&= \mathrm{Cov}(Z, e_Y).
\end{aligned}
$$

$\square$

We learn a model for $\hat{Y} = \mathbb{E}[Y | X, T]$ and compute its residuals as $Y - \mathbb{E}[Y | X, T]$. Notice $Y - \mathbb{E}[Y | X, T] = Y - \varphi(X, T) - (E[Y | X, T] - \varphi(X, T)) = e_Y - \mathbb{E}[e_Y | X, T]$. Lemma 1 suggests how to construct of a loss term which enforces unconfoundedness. Notice that regardless of whether any existing instruments are normally distributed, if we can construct $g(X) = Z \sim \mathcal{N}(0, \sigma^2)$ to have no covariance with the residuals $e_Y - \mathbb{E}[e_Y | X, T]$, i.e. with $Y - \mathbb{E}[Y | X, T]$, then $Z$ has zero covariance with $e_Y$ by Lemma 1. Together, this suggests the following model constraints:

**Constraint 1** *(Instrumental Unconfoundedness)*: $\mathrm{Cov}(g(X), e_Y) = 0$, $Z \sim \mathcal{N}(0, \sigma^2)$.

**Constraint 2** *(Exclusion Restriction)*: $\mathrm{Cov}(f(X), Y) > 0$, $\mathrm{Cov}(g(X), f(X)) = 0$.

**Constraint 3** *(Relevance)*: $\mathrm{Cov}(T, g(X)) > 0$.

## 4 RELATED WORK

The majority of existing works for learning IVs automate IV selection from observed candidates. Meaning these works recover existing IVs. ModeIV chooses instruments by looking at clusters of treatment effects based on weighting the observed variables used as instruments (Hartford et al., 2021). DIV.VAE uses a variational autoencoder (VAE) approach to disentangle an instrument under the assumption that a surrogate instrument exists in the data (Cheng et al., 2024). IV.Tetrad (Silva & Shimizu, 2017) builds strong instruments requiring at least two observed IV candidates. Several methods for refining IV candidates and estimating causal effects from these candidates can fall into

this category as well including sisVIVE (Kang et al., 2016), TEDVAE (Zhang et al., 2021) and Ivy (Kuang et al., 2020) among many others (Davies et al., 2015; Burgess et al., 2016).

Methods to learn IVs were also proposed in the GIV (Wu et al., 2023), AutoIV (Yuan et al., 2022), VIV (Li & Yao, 2024), DVAE.CIV (Cheng et al., 2023), and GDIV (Chou et al., 2024). GIV generates a categorical IV with unsupervised expectation-maximization that groups data according to underlying distributional differences assumed to arise from the aggregation of data from multiple sources. This automates the idea of using environment as an IV in data coming multiple sources (Schweisthal et al., 2024). The other methods learn variational distributions. The AutoIV method uses a mutual information (MI) based loss to generate an abstract IV from observed data by learning variational distributions (Yuan et al., 2022). VIV, DVAE.CIV, and GDIV use VAEs to learn independent latent variables that serve as $Z, U, C$ from the observed data $Y, T, X$, sometimes including an additional adjustment variable $A$ derived from $Y, X$. VAEs have shown great success in probabilistic modeling in general but lack theory to guarantee learning the true causal model and satisfaction of IV conditions.

## 5 METHODS

We introduce an architecture which we call ZNet, a multi-armed multi-loss network (Figure 3) specifically constructed to enforce Constraints 1-3 and learn a new SCM. The network contains four feed forward neural networks, $\Phi, f, g, \pi$: $\Phi$ is our model for $\mathbb{E}[Y|X,T]$, $f, g$ learn to derive the instrument $Z$ and confounders $C$ from $X$, and $\pi$ estimates the treatment $T$ from the derived instrument $Z$. The losses force networks $f, g$ to learn latent representations from the input observational dataset $\{X, T, Y\}$ leveraging $\Phi, \pi$. The networks $f, g, \pi, \Phi$ each consist

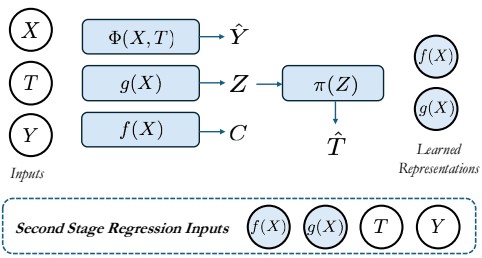

Figure 3: **ZNet Architecture.**

of two hidden layers, where the activation function can be chosen between ReLU or linear. The output layers of $f, g$ may use either a linear mapping or a temperature-scaled softmax. From our trained ZNet, we learn representations for $\{C, Z\}$ from $X$ and use the existing $T, Y$ to assemble the dataset $\{C, Z, T, Y\}$. We use downstream IV estimators on this data to predict and evaluate treatment effects across data settings. We compare treatment effect estimation using ZNet to GIV, AutoIV, and VIV.

### 5.1 ZNET LOSS TERMS

The ZNet multi-part loss function automates IV generation by forcing Constraints 1-3. We first consider a Pearson correlation-based (PC) loss, which between any two variables $A$ and $B$ is the ratio of the covariance between $A$ and $B$ and the product of the standard deviations of each of $A$ and $B$:

$$PC(A, B) = \frac{\text{Cov}(A, B)}{\sigma_A \sigma_B}. \tag{4}$$

Each time we seek to minimize covariance, we minimize $(PC)^2$. In contrast, for relationships we seek to maximize covariance, we minimize $1 - (PC)^2$. To increase the generality of ZNet beyond linear settings, we additionally employ a mutual information (MI) based loss, minimized or maximized in the same respective manner, and approximated using kernel density estimation (KDE) with Guassian kernels.

**Enforcing Constraint 1** *(Instrumental Unconfoundedness)*: To enforce unconfoundedness, we leverage Lemma 1. We first learn a model $\Phi$ to predict $\hat{Y} = \Phi(X \odot T)$ from $X$ and $T$ using mean squared error (MSE). Then via a loss minimizing the correlation between the error $Y - \hat{Y}$ and $Z$, we encourage zero correlation of $Z$ with $e_Y$:

$$L_{X,T \to Y} = \alpha_1 \cdot MSE(\Phi(X \odot T), Y) \tag{5}$$

$$L_{Z \not\to e_Y}^{PC} = \alpha_2 \cdot PC(Y - \hat{Y}, Z)^2 \tag{6}$$

As $L^{PC}_{Z \not\to e_Y}$ approaches 0, satisfaction of Constraint 1 and thereby instrumental unconfoundedness is reached. The loss $L_{X,T \to Y}$ is trained separately and first.

**Enforcing Constraint 2** (*Exclusion Restriction*): We need $C$ to capture all observed variation in $Y$ not through $T$. We discourage $Z$ from being directly predictive of $Y$, i.e. remove additional information about $Y$ conditional on $C$ and $T$, by encouraging $C$ to be highly correlated with $Y$ and $Z$ to have zero covariance with $C$, i.e. combining the following two losses encourages Constraint 2 and exclusion restriction by preventing $Z$ from entering $\varphi'$:

$$L^{PC}_{C \to Y} = \alpha_3 \cdot \left(1 - PC(C,Y)^2\right) + \alpha_4 \cdot MSE(C,Y) \tag{7}$$

$$L^{PC}_{Z \perp C} = \alpha_5 \cdot PC(C,Z)^2 \tag{8}$$

**Enforcing Constraint 3** *(Relevance):* We enforce relevance of the learned instrument $Z$ to the treatment variable $T$ by forcing its predictive power and correlation. When $T$ is binary, we have

$$L^{PC}_{Z \to T}(Z) = \alpha_6 \cdot BCE(\pi(Z),T) + \alpha_7 \cdot (1 - PC(Z,T)^2) \tag{9}$$

using binary cross-entropy (BCE). We would replace BCE by mean squared error (MSE) when the treatment $T$ is continuous.

$Z$ **and** $C$ **Distribution Losses** We use a Kullback-Leibler (KL) divergence loss on each dimension of $Z$ and $C$ with a mean-zero normal distribution to stabilize $Z, C$ and to force learning $Z \sim \mathcal{N}(0, \sigma^2)$ for the sake of Lemma 1. We also minimize the average PC across dimensions within $C$ and $Z$ to encourage the features of the learned representations to be distinct.

## 5.2 TREATMENT EFFECT ESTIMATION

We use three downstream estimators of treatment effects to demonstrate the ability of ZNet for causal inference: TSLS, DFIV and DeepIV. Each method takes as input the true treatment term $T$ and our learned representations $C$ and $Z$. TSLS is the classical IV estimator. It assumes linear structural equations and independence of $U$ and $X$ (Imbens & Rubin, 2015; Verbeek, 2004). DeepIV (Hartford et al., 2017) generalizes TSLS by allowing the model at each stage to be parameterized by a neural network and $X \not\perp U$. The DFIV estimator is a second leading estimator which allows basis functions at each stage to be parametrized by neural networks (Xu et al., 2020). We compare our pipeline to using the true instrument *TrueIV*, if it exists, and to TARNet (Shalit et al., 2017), a state of the art treatment effect estimator used in the absence of an IV.

## 5.3 TRAINING

ZNet training occurs in three stages. First, $\Phi$ is trained to predict $Y$ from $X$ and $T$ using the MSE loss $L_{X,T \to Y}$, i.e. only $\alpha_1$ is non-zero. The $\Phi$ network is then frozen. Next $f, g$ are pretrained with all loss coefficients set to 0 except for $\alpha_3, \alpha_6$ to encourage a starting representation for $C$ relevant to $Y$ and $Z$ relevant to $T$. Then ZNet $(f, g, \pi)$ is trained with the full loss to learn the SCM with $\{Z, C, T, Y\}$. In training ZNet, our loss terms are potentially conflicting, so to stabilize training, we allowed the network to use gradient surgery (Yu et al., 2020).

Hyperparameters, including loss term weights, whether constraints are PC or MI, and the necessity of gradient surgery, were tuned using Bayesian optimization implemented in Botorch (Balandat et al., 2020). We perform the optimization in two stages. For each IV generation method (ZNet, AutoIV, GIV, and VIV), we maximized the instrument's relevance F-Statistic and minimized the correlation between learned $C$ and $Z$ using Botorch's native adaptation of the Noisy Expected Improvement acquisition function for multi-objective optimization. We then choose the parameter set from the Pareto front with the highest F-Statistic. We tune the causal inference methods (DeepIV, DFIV, and TARNet) to simultaneously minimize the MSE of the model's ATE against a nearest-neighbors (NN) ATE and the MSE of estimated Y on factual Y, again with the Noisy Expected Improvement acquisition function. The parameter set is selected from the Pareto front by least NN ATE MSE.

# 6 EVALUATION

## 6.1 DATA GENERATION

For evaluation, we focus on binary treatments, though ZNet could easily be adapted for continuous settings. We construct multiple semi-synthetic datasets to evaluate ZNet's ability to predict causal effects across settings. The **IHDP** data is a common causal inference benchmark dataset (Hill, 2011). It is data based on an experiment that studied the effect of home visits during infancy on cognitive test scores of premature infants. There are 985 individuals and 25 covariates. We build our data from these covariates, masking some covariates to serve as unobserved confounding. We define three sets of covariates, $X^{\rightarrow T}$, $X^{\rightarrow Y}$, and $X^{\leftarrow U}$, where we each $X^{\rightarrow I}$ is the subset of covariates $X$ which have a causal relationship with the covariate subset $I$ in the arrow's direction. We create the following classes of data based on their inclusion of an instrument:

1. **Disjoint Candidate:** $\exists\ X^{\rightarrow T}\ s.t.\ X^{\rightarrow T} \cap X^{\rightarrow Y} = \varnothing, X^{\rightarrow T} \cap X^{\leftarrow U} = \varnothing$

2. **Mixed Candidate:** $\exists\ \widetilde{X}^{\rightarrow T} \subset X^{\rightarrow T}\ s.t.\ \widetilde{X}^{\rightarrow T} \cap X^{\rightarrow Y} = \varnothing, \widetilde{X}^{\rightarrow T} \cap X^{\leftarrow U} = \varnothing$

3. **Latent Categorical Instrument:** $\exists\ Z, f\ s.t.\ f(X^{\rightarrow T}) = Z \in \mathbb{N}^+$

4. **No Candidate** $\nexists\ \widetilde{X}^{\rightarrow T} \subseteq X^{\rightarrow T}\ s.t.\ \widetilde{X}^{\rightarrow T} \cap X^{\rightarrow Y} = \varnothing, \widetilde{X}^{\rightarrow T} \cap X^{\leftarrow U} = \varnothing$

For each class, we consider $X^{\leftarrow U} \neq \varnothing$ and, in the appendix, $X^{\leftarrow U} = \varnothing$. We also consider data where $U = \varnothing$ (i.e. no unobserved confounding). After fixing covariate sets, we choose functions $\phi, \psi, e_Y, e_T$ and generate the variables $Y, T$ similar to (Wu et al., 2023) by writing

$$Y = \phi(X_Y, T) + e_Y(U) + \varepsilon_Y \text{ for } \varepsilon_Y \sim \mathcal{N}(0, .1) \tag{10}$$

$$T \sim \text{Bernoulli}(P) \text{ for } P = \psi(X_T) + e_T(U) + \varepsilon_T \text{ for } \varepsilon_T \sim \mathcal{N}(0, .1). \tag{11}$$

We consider a linear and non-linear version of $\phi, \psi$ for each dataset. Data are split into 60% for training, 20% for validation, and 20% for testing. All experimental results are that of the test data.

## 6.2 LEARNING INSTRUMENTS WITH ZNET

ZNet successfully recovers existing instruments. In the **Linear Mixed Candidate** dataset, there are three variables $X13, X14, X15 \in X^{\rightarrow T}$ which are instruments. ZNet chooses to generate a 10-dimensional variable $Z$ which is correlated with and linearly predicts each of $X13, X14, X15$ (Figure 5 a,b). Instrument recovery is due to the combination of ZNet loss constraints. Upon ablation of each, recovery deteriorates. We see this in the decreasing ability to predict the true instruments from that recovered by the network without each component (Figure 5 c). We see similar performance in other datasets with candidates and include a non-linear example in Appendix Figure 7.

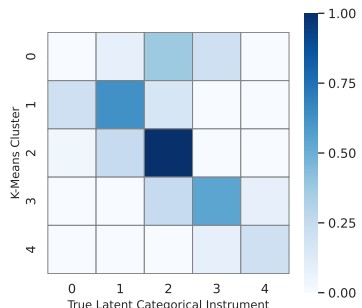

Figure 4: **Normalized confusion matrix demonstrating ZNet recovery of linear latent categorical instrument.**

ZNet is also able to recover latent instruments. We demonstrate this with our **Linear Categorical Instrument** dataset. The true instrument groups the observed data into 5 clusters. ZNet can be seen to approximately recover these clusters after K-Means and cluster relabeling in Figure 4.

Independent of the existence of an instrument in the observed data, ZNet generates an instrument representation that is correlated with $T$, independent of the confounder representation $C$, independent of the error in predicting $Y$, and unconfounded by $U$. We evaluate the suitability of this instrument representation empirically. We demonstrate this with our **Non-linear No Candidate** dataset. The generated instrument representation is relevant to $T$, not additionally helpful in predicting $Y$, and shows weak correlation to unobserved confounders (Figure 6). We observe strong F-Statistics for $T$ prediction from generated representations $Z$ and low PC across prohibited relationships between $Z$ and confounders in the other datasets as well which we report in Appendix Tables 7, 8.

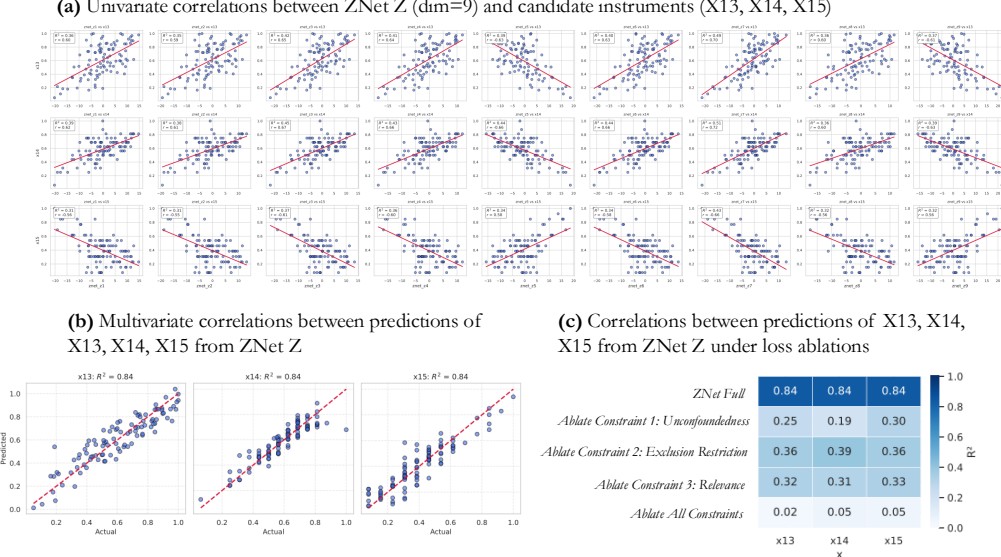

Figure 5: **Learned instrument representation is correlated to existing instruments in linear dataset with mixed instrument candidate.** a) Learned instruments scattered against the true instruments. b) Regression predictions from learned $Z$ dimensions predicting the true instruments scattered against the true instruments. c) Regression $R^2$ values for predicting the true instrument with ZNet learned instruments across loss ablation experiments.

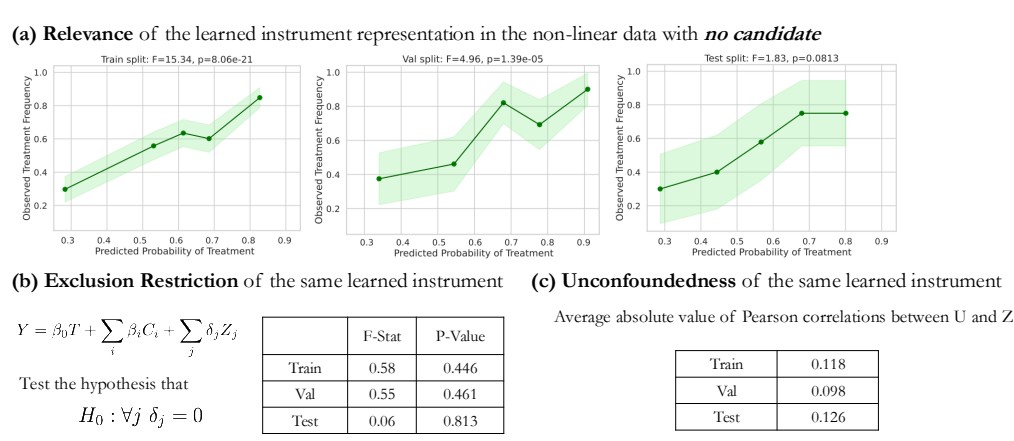

Figure 6: **Learned instrument representation is valid even in the absence of real instruments in nonlinear data with no instrument candidate.** a) We see learned instruments are *relevant* based on calibration plots of regression of $T$ on learned $Z$. b) Exclusion restriction is satisfied as representations $Z$ do not improve the prediction of $Y$ after accounting for the treatment and learned confounders $C$ (F-tests are not significant). c) Learned instrument representations $Z$ show minimal correlation with the unobserved confounders $U$.

## 6.3 CAUSAL INFERENCE WITH ZNET LEARNED REPRESENTATIONS

ZNet learned representations, along with those of AutoIV, GIV, and VIV, can recover ATE and CATE after a second stage regression, i.e. TSLS, DeepIV, or DFIV. Performance of ZNet is comparable to using the ground truth instrument, TrueIV, when available, and IV generation generally exceeds that of TARNet, which ignores confounding, for both ATE, Table 1, and CATE, Appendix Tables 3, 4. ZNet is on average the highest performing among IV generation methods across comprehensive data generation processes (Appendix Tables 9, 10). Notably, in the setting of no unobserved confounding (no $U$) without a candidate instrument, ZNet is comparable to TARNet. Given that we cannot

| Dataset | Diff Means | TARNet | IV Method | TSLS | DeepIV | DF IV |
|---|---|---|---|---|---|---|
| Linear Disjoint
True ATE: 0.815 | 0.054 | -0.025 | TrueIV
ZNet
AutoIV
VIV
GIV | **-0.002**$^{**}$
*0.119*
-1.393
0.147
-0.620 | 0.108
*0.054*$^*$
**0.038**$^{**}$
0.123
0.115 | **0.132**$^{**}$
*-0.303*$^*$
-0.964
0.546
0.304 |
| Linear Latent
True ATE: 0.941 | -0.539 | -0.146 | TrueIV
ZNet
AutoIV
VIV
GIV | -0.524
*-0.125*
-1.315
**-0.082**
0.285 | -0.317
**-0.136**$^{**}$
-0.309
*-0.171*$^*$
-0.234 | **0.042**$^{**}$
-0.231
-0.270
*-0.122*$^*$
-0.447 |
| Linear Mixed
True ATE: 0.608 | 0.407 | 0.429 | TrueIV
ZNet
AutoIV
VIV
GIV | **0.263**$^{**}$
*0.437*
-0.803
1.349
1.171 | *0.429*$^*$
**0.381**$^{**}$
0.548
0.637
0.525 | 0.369
0.655
0.270
*-0.256*$^*$
**0.217**$^{**}$ |
| Linear No Candidate (no $U$)
True ATE: 1.882 | -0.296 | -0.169 | TrueIV
ZNet
AutoIV
VIV
GIV | –
2.718
0.963
*0.279*
**0.137** | –
*-0.033*$^*$
**-0.017**$^{**}$
-0.111
-0.097 | –
-0.336
*-0.300*$^*$
**-0.107**$^{**}$
-0.741 |
| Linear No Candidate
True ATE: 0.354 | 0.657 | 0.240 | TrueIV
ZNet
AutoIV
VIV
GIV | –
**0.025**
*-0.028*
0.305
-2.614 | –
*0.189*$^*$
0.251
**0.185**$^*$
0.278 | –
*0.156*$^*$
0.565
0.632
**-0.031**$^{**}$ |
| Non-linear Disjoint
True ATE: 0.544 | 0.766 | 0.324 | TrueIV
ZNet
AutoIV
VIV
GIV | **0.266**$^{**}$
*0.524*
1.511
0.561
0.697 | **0.272**$^{**}$
*0.309*$^*$
0.389
0.555
0.365 | **-0.103**$^{**}$
*0.147*$^*$
-0.403
-0.214
1.120 |
| Non-linear Latent
True ATE: 0.333 | 0.528 | 0.050 | TrueIV
ZNet
AutoIV
VIV
GIV | 1.381
**0.152**
-4.809
1.790
*-0.235* | *-0.020*
-0.039
**-0.008**$^{**}$
-0.039
-0.028 | 4.762
**-0.063**$^{**}$
0.785
-0.170
*0.084*$^*$ |
| Non-linear Mixed
True ATE: 0.558 | 0.849 | 0.255 | TrueIV
ZNet
AutoIV
VIV
GIV | *0.477*
**0.244**$^*$
10.821
0.950
-0.981 | *0.142*$^*$
0.218
**0.036**$^{**}$
0.408
0.293 | *-0.156*$^*$
**0.033**$^{**}$
2.079
0.847
0.983 |
| Non-linear No Candidate (no $U$)
True ATE: 1.429 | 0.250 | -0.068 | TrueIV
ZNet
AutoIV
VIV
GIV | –
*-0.528*
-1.806
**0.182**$^{**}$
3.389 | –
**-0.012**$^{**}$
0.064
-0.085
*0.053* | –
**-0.143**$^{**}$
-0.257
*-0.209*$^*$
-0.665 |
| Non-linear No Candidate
True ATE: 0.435 | 0.783 | 0.423 | TrueIV
ZNet
AutoIV
VIV
GIV | –
*0.200*$^*$
-25.181
0.898
**-0.109** | –
**0.260**$^{**}$
0.720
*0.422*$^*$
0.640 | –
**0.049**$^{**}$
0.477
0.404
*0.345*$^*$ |

Table 1: **Mean error on ATE by dataset and causal inference method across 50 resampled bootstraps.** Smallest errors are **bolded**. Second smallest are *italicized*. A single * indicates that the two best are significantly better than the third best. Two ** indicates that the best is significantly better than the second best.

assess the existence (or lack thereof) of unobserved confounding in non-synthetic datasets, ZNet's performance on these datasets support its translation to real-world settings.

# 7 DISCUSSION

We present novel methodology for data driven learning of IV representations using deep learning with superior performance. Our network, ZNet, differs from existing literature generating IVs in its approach. Existing methods learn variational distributions, while our method learns SCMs. Existing methods assume that unobserved confounders do not influence the observed data, while our method

relaxes this assumption. These make our implementation simple and transparent for widespread utility. We demonstrate that ZNet is able to recover valid instrument representations. In the case of existing instruments among the observed data, recovered instruments are highly correlated with these variables. This is shown empirically in cases when the instrument was either observed or latent. Regardless of the existence of instruments in the data, ZNet shows strong performance predicting treatment effects across settings of unobserved confounding performing on average better than existing variational methods for IV generation.

ZNet eliminates the need for domain knowledge of pre-existing IVs by automating instrument representations from observed data. We contribute the most comprehensive evaluation of IV generation for causal inference, which demonstrates the broad utility of IV generation. We present performance across a comprehensive collection of data generation settings. Since the data generation process is untestable in practice, these results suggest that ZNet can serve as a plug-in causal inference estimator. ZNet is high performing across these semi-synthetic settings. Regardless of the existence of a candidate or a latent instrument, or of unobserved confounding, ZNet can match or exceed the performance of TARNet and of probabilistic IV generation methods.

Solutions to the ZNet loss minimization problem will always give a representation that serves as an instrument since IV constraints are explicitly embedded in the loss function. This instrument can then be used in any downstream instrument regression where satisfying the standard IV criteria (or, equivalently, ZNet criteria) implies the validity of subsequent causal inference. However, IV estimation in general is limited by a lack of theoretical guarantees of identifiably in the general case. This theoretically limits our approach and IV estimation in general. However, strong empirical results alongside ongoing work to stabilize downstream IV estimators, i.e. (Li et al., 2024), suggest the value in the increased use of these methods beyond linear settings. We see great potential for IV estimation in general and our methods in particular with the growing use of unstructured data. Unstructured data may contain latent or abstract instruments more frequently, as high-dimensional feature spaces often contain rich information that our approach could learn to extract as instruments. Our method's simplicity adds interpretability. Learning SCMs through constraints allows for direct control over the strength and validity of learned instruments, which elucidates performance in the absence of theoretical guarantees on downstream causal inference. Due to its lack of assumptions on the data generation process, ZNet suggests that IV generation presents the potential to strengthen causal inference and broaden its applicability.

## ETHICS STATEMENT

There are no privacy, fairness, security, or other ethics concerns with this work. Large language models were used for assisting in code production (i.e. aide in plotting results, converting code-bases for comparison from TensorFlow to PyTorch, implementing MI approximation and developing Bayesian tuning) and literature review.

## REPRODUCIBILITY STATEMENT

We make our results reproducible by providing all details of their construction and associated assumptions in the paper. Moreover, all code to generate models, synthetic data, and experiments will be made public upon publication.

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

# A  APPENDIX / SUPPLEMENTAL MATERIAL

## A.1  ADDITIONAL FIGURES

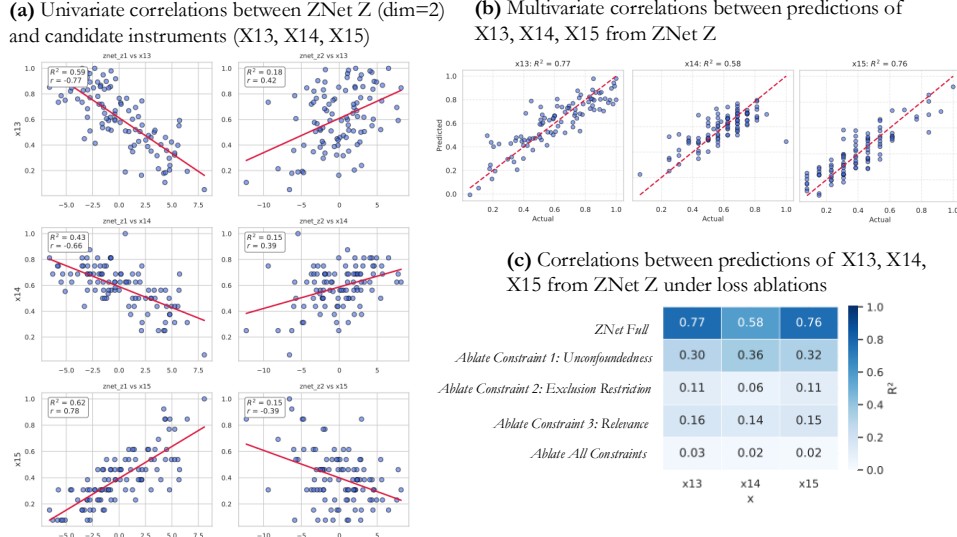

Figure 7: **Learned instrument representation is correlated to existing instruments in non-linear dataset with mixed instrument candidate in test set.** a) Learned instruments against the true instruments. b) Regression predictions from learned $Z$ dimensions predicting the true instruments. c) Regression $R^2$ values for predicting the true instrument with ZNet learned instruments across loss ablation experiments.

| Dataset | Method | X13 | X14 | X15 |
|---|---|---|---|---|
| Linear Disjoint | ZNet Full | 0.593 | 0.542 | 0.691 |
| | Ablate Unconfoundedness Constraint | 0.629 | 0.435 | 0.705 |
| | Ablate Exclusion Restriction Constraint | 0.666 | 0.554 | 0.749 |
| | Ablate Relevance Constraint | 0.586 | 0.440 | 0.542 |
| | Ablate All Constraint | 0.016 | 0.023 | 0.032 |
| Linear Disjoint (no $U \to X$) | ZNet Full | 0.784 | 0.707 | 0.811 |
| | Ablate Unconfoundedness Constraint | 0.616 | 0.453 | 0.604 |
| | Ablate Exclusion Restriction Constraint | 0.714 | 0.514 | 0.682 |
| | Ablate Relevance Constraints | 0.271 | 0.207 | 0.367 |
| | Ablate All Constraints | 0.259 | 0.143 | 0.124 |
| Linear Mixed | ZNet Full | 0.837 | 0.835 | 0.838 |
| | Ablate Unconfoundedness Constraint | 0.255 | 0.194 | 0.302 |
| | Ablate Exclusion Restriction Constraint | 0.355 | 0.392 | 0.358 |
| | Ablate Relevance Constraint | 0.322 | 0.311 | 0.329 |
| | Ablate All Constraints | 0.024 | 0.054 | 0.050 |
| Linear Mixed (no $U \to X$) | ZNet Full | 0.711 | 0.624 | 0.591 |
| | Ablate Unconfoundedness Constraint | 0.259 | 0.353 | 0.228 |
| | Ablate Exclusion Restriction Constraint | 0.306 | 0.359 | 0.411 |
| | Ablate All Constraints | 0.023 | 0.018 | 0.033 |
| Non-linear Disjoint | ZNet Full | 0.361 | 0.387 | 0.293 |
| | Ablate Unconfoundedness Constraint | 0.181 | 0.202 | 0.283 |
| | Ablate Exclusion Restriction Constraint | 0.410 | 0.423 | 0.442 |
| | Ablate Relevance Constraint | 0.220 | 0.213 | 0.239 |
| | Ablate All Constraints | 0.092 | 0.058 | 0.033 |
| Non-linear Disjoint (no $U \to X$) | ZNet Full | 0.532 | 0.384 | 0.516 |
| | Ablate Unconfoundedness Constraint | 0.285 | 0.175 | 0.282 |
| | Ablate Exclusion Restriction Constraint | 0.372 | 0.414 | 0.421 |
| | Ablate Relevance Constraint | 0.009 | 0.035 | 0.004 |
| | Ablate All Constraints | 0.068 | 0.024 | 0.045 |
| Non-linear Mixed | ZNet Full | 0.767 | 0.577 | 0.759 |
| | Ablate Unconfoundedness Constraint | 0.299 | 0.357 | 0.321 |
| | Ablate Exclusion Restriction Constraint | 0.109 | 0.064 | 0.106 |
| | Ablate Relevance Constraint | 0.164 | 0.137 | 0.154 |
| | Ablate All Constraints | 0.028 | 0.020 | 0.017 |
| Non-linear Mixed (no $U \to X$) | ZNet Full | 0.209 | 0.120 | 0.178 |
| | Ablate Unconfoundedness Constraint | 0.463 | 0.273 | 0.387 |
| | Ablate Exclusion Restriction Constraint | 0.362 | 0.242 | 0.415 |
| | Ablate Relevance Constraint | 0.369 | 0.272 | 0.421 |
| | Ablate All Constraints | 0.027 | 0.060 | 0.036 |

Table 2: **Multivariate $R^2$ for recovering instruments $X13, X14, X15$ for each dataset and method**.

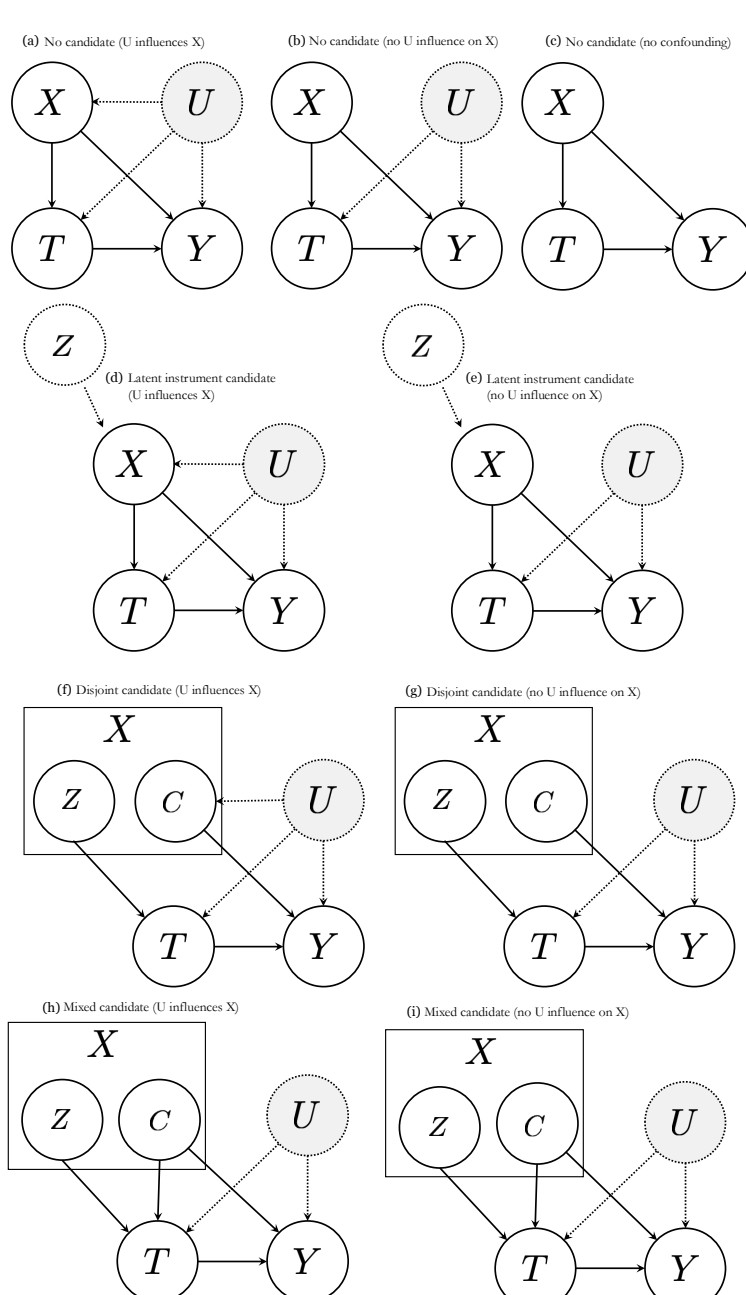

Figure 8: **Directed acyclic graphs (DAGs) demonstrating the various data generation processes on which ZNet is evaluated**. Linear and non-linear relationships are constructed for each DAG giving 18 total datasets for evaluation. Maintext results focus on cases where $U$ influences $X$ as this is more challenging, more general, and unique to ZNet.

| Dataset | TARNet | IV Method | DeepIV | DF IV |
|---|---|---|---|---|
| Linear Disjoint | 0.103 | TrueIV | 0.223 | **0.180** |
| | | ZNet | *0.165* | *0.315* |
| | | AutoIV | **0.113** | 0.993 |
| | | VIV | 0.236 | 0.593 |
| | | GIV | 0.209 | 0.542 |
| Linear Disjoint (no $U \to X$) | 0.410 | TrueIV | *0.394* | **0.306** |
| | | ZNet | **0.329** | 0.500 |
| | | AutoIV | 0.432 | 1.541 |
| | | VIV | 0.439 | *0.328* |
| | | GIV | 0.499 | 0.363 |
| Linear Latent | 0.170 | TrueIV | 0.364 | **0.142** |
| | | ZNet | *0.267* | 0.236 |
| | | AutoIV | 0.367 | 0.291 |
| | | VIV | **0.260** | *0.193* |
| | | GIV | 0.324 | 0.471 |
| Linear Latent (no $U \to X$) | 0.474 | TrueIV | *0.498* | **0.153** |
| | | ZNet | **0.472** | 0.651 |
| | | AutoIV | 0.520 | *0.328* |
| | | VIV | 0.538 | 0.372 |
| | | GIV | 0.518 | 0.861 |
| Linear Mixed | 0.435 | TrueIV | *0.541* | 0.418 |
| | | ZNet | **0.459** | 2.182 |
| | | AutoIV | 0.664 | 0.398 |
| | | VIV | 0.746 | *0.264* |
| | | GIV | 0.649 | **0.246** |
| Linear Mixed (no $U \to X$) | 0.403 | TrueIV | 0.500 | 0.781 |
| | | ZNet | **0.344** | *0.375* |
| | | AutoIV | 0.793 | 2.337 |
| | | VIV | 0.561 | 0.648 |
| | | GIV | *0.493* | **0.237** |
| Linear No Candidate | 0.278 | TrueIV | – | – |
| | | ZNet | 0.471 | *0.199* |
| | | AutoIV | *0.357* | 0.581 |
| | | VIV | 0.389 | 0.698 |
| | | GIV | **0.336** | **0.130** |
| Linear No Candidate (no $U \to X$) | 0.425 | TrueIV | – | – |
| | | ZNet | *0.557* | *0.318* |
| | | AutoIV | 0.569 | **0.272** |
| | | VIV | 0.666 | 0.427 |
| | | GIV | **0.471** | 0.374 |
| Linear No Candidate (no $U$) | 0.193 | TrueIV | – | – |
| | | ZNet | **0.173** | 0.431 |
| | | AutoIV | 0.353 | *0.401* |
| | | VIV | *0.236* | **0.236** |
| | | GIV | 0.301 | 0.761 |

Table 3: **PEHE on linear synthetic datasets**.

| Dataset | TARNet | IV Method | DeepIV | DF IV |
|---|---|---|---|---|
| Non-linear Disjoint | 0.531 | TrueIV | *0.539* | 0.383 |
| | | ZNet | **0.467** | *0.332* |
| | | AutoIV | 0.585 | 0.467 |
| | | VIV | 0.727 | **0.325** |
| | | GIV | 0.550 | 1.140 |
| Non-linear Disjoint (no $U \to X$) | 1.158 | TrueIV | **0.560** | 2.098 |
| | | ZNet | 0.748 | **0.656** |
| | | AutoIV | *0.577* | 1.262 |
| | | VIV | 0.624 | *0.727* |
| | | GIV | 0.593 | 5.709 |
| Non-linear Latent | 0.108 | TrueIV | 0.261 | 4.784 |
| | | ZNet | **0.198** | **0.162** |
| | | AutoIV | *0.253* | 0.866 |
| | | VIV | 0.370 | 0.348 |
| | | GIV | 0.259 | *0.197* |
| Non-linear Latent (no $U \to X$) | 0.452 | TrueIV | 0.355 | **0.218** |
| | | ZNet | *0.264* | *0.451* |
| | | AutoIV | 0.459 | 0.796 |
| | | VIV | 0.572 | 0.459 |
| | | GIV | **0.261** | 2.703 |
| Non-linear Mixed | 0.346 | TrueIV | **0.326** | *0.281* |
| | | ZNet | *0.362* | **0.242** |
| | | AutoIV | 0.446 | 2.264 |
| | | VIV | 0.694 | 0.874 |
| | | GIV | 0.439 | 0.992 |
| Non-linear Mixed (no $U \to X$) | 0.756 | TrueIV | 0.847 | 0.976 |
| | | ZNet | **0.665** | **0.669** |
| | | AutoIV | 0.975 | 0.799 |
| | | VIV | 0.906 | *0.698* |
| | | GIV | *0.839* | 1.069 |
| Non-linear No Candidate | 0.562 | TrueIV | – | – |
| | | ZNet | **0.402** | **0.423** |
| | | AutoIV | 0.788 | 0.611 |
| | | VIV | *0.588* | 0.476 |
| | | GIV | 0.712 | *0.429* |
| Non-linear No Candidate (no $U \to X$) | 0.681 | TrueIV | – | – |
| | | ZNet | 0.960 | 0.667 |
| | | AutoIV | **0.679** | *0.586* |
| | | VIV | 0.776 | **0.551** |
| | | GIV | *0.716* | 0.604 |
| Non-linear No Candidate (no $U$) | 1.148 | TrueIV | – | – |
| | | ZNet | *1.157* | 1.124 |
| | | AutoIV | **1.138** | **1.056** |
| | | VIV | 1.227 | *1.082* |
| | | GIV | 1.179 | 1.152 |

Table 4: **PEHE on non-linear synthetic datasets**.

| Dataset | Diff Means | TARNet | IV Method | TSLS | DeepIV | DF IV |
|---|---|---|---|---|---|---|
| Linear Disjoint (no $U \rightarrow X$)
True ATE: 0.745 | -0.327 | -0.311 | TrueIV | *0.266** | **-0.264**** | *-0.271** |
| | | | ZNet | -0.356 | *-0.295** | -0.469 |
| | | | AutoIV | **-0.044** | -0.361 | -1.454 |
| | | | VIV | 0.704 | -0.353 | **-0.074**** |
| | | | GIV | -1.456 | -0.426 | -0.304 |
| Linear Disjoint
True ATE: 0.815 | 0.054 | -0.025 | TrueIV | **-0.002**** | 0.108 | **0.132**** |
| | | | ZNet | *0.119* | *0.054** | *-0.303** |
| | | | AutoIV | -1.393 | **0.038**** | -0.964 |
| | | | VIV | 0.147 | 0.123 | 0.546 |
| | | | GIV | -0.620 | 0.115 | 0.304 |
| Linear Latent (no $U \rightarrow X$)
True ATE: 0.957 | -0.498 | -0.427 | TrueIV | **-0.171** | -0.445 | **0.068**** |
| | | | ZNet | -1.577 | *-0.406** | -0.506 |
| | | | AutoIV | 1.505 | -0.462 | *-0.221** |
| | | | VIV | *-0.210* | -0.476 | -0.308 |
| | | | GIV | 0.245 | **-0.372**** | -0.864 |
| Linear Latent
True ATE: 0.941 | -0.539 | -0.146 | TrueIV | -0.524 | -0.317 | **0.042**** |
| | | | ZNet | *-0.125* | **-0.136**** | -0.231 |
| | | | AutoIV | -1.315 | -0.309 | -0.270 |
| | | | VIV | **-0.082** | *-0.171** | *-0.122** |
| | | | GIV | 0.285 | -0.234 | -0.447 |
| Linear Mixed (no $U \rightarrow X$)
True ATE: 1.569 | -0.396 | -0.297 | TrueIV | *-0.260** | -0.304 | -0.690 |
| | | | ZNet | **-0.112*** | **0.005**** | *-0.165** |
| | | | AutoIV | 3.508 | 0.226 | -1.906 |
| | | | VIV | 0.883 | *-0.100** | -0.581 |
| | | | GIV | 2.462 | -0.461 | **-0.072**** |
| Linear Mixed
True ATE: 0.608 | 0.407 | 0.429 | TrueIV | **0.263**** | *0.429** | 0.369 |
| | | | ZNet | *0.437* | **0.381**** | 0.655 |
| | | | AutoIV | -0.803 | 0.548 | 0.270 |
| | | | VIV | 1.349 | 0.637 | *-0.256** |
| | | | GIV | 1.171 | 0.525 | **0.217**** |
| Linear No Candidate (no $U \rightarrow X$)
True ATE: 0.952 | -0.062 | -0.085 | TrueIV | – | – | – |
| | | | ZNet | -0.630 | **-0.088*** | -0.214 |
| | | | AutoIV | -2.496 | -0.110 | **0.050**** |
| | | | VIV | *0.404* | *0.016** | *0.337** |
| | | | GIV | **0.372** | -0.175 | -0.353 |
| Linear No Candidate (no $U$)
True ATE: 1.882 | -0.296 | -0.169 | TrueIV | – | – | – |
| | | | ZNet | 2.718 | *-0.033** | -0.336 |
| | | | AutoIV | 0.963 | **-0.017**** | *-0.300** |
| | | | VIV | *0.279* | -0.111 | **-0.107**** |
| | | | GIV | **0.137** | -0.097 | -0.741 |
| Linear No Candidate
True ATE: 0.354 | 0.657 | 0.240 | TrueIV | – | – | – |
| | | | ZNet | **0.025** | *0.189** | *0.156** |
| | | | AutoIV | *-0.028* | 0.251 | 0.565 |
| | | | VIV | 0.305 | **0.185*** | 0.632 |
| | | | GIV | -2.614 | 0.278 | **-0.031**** |

Table 5: **ATE results on synthetic linear datasets**.

| Dataset | Diff Means | TARNet | IV Method | TSLS | DeepIV | DF IV |
|---|---|---|---|---|---|---|
| Non-linear Disjoint (no $U \to X$)
True ATE: 0.919 | -0.481 | -0.372 | TrueIV
ZNet
AutoIV
VIV
GIV | *-0.864*
**-0.514**
-1.676
-4.066
-25.776 | *-0.207*\*
-0.635
-0.410
-0.225
**-0.196**\* | 2.097
**-0.534**\*\*
-1.180
*-0.608*\*
-5.724 |
| Non-linear Disjoint
True ATE: 0.544 | 0.766 | 0.324 | TrueIV
ZNet
AutoIV
VIV
GIV | **0.266**\*\*
*0.524*
1.511
0.561
0.697 | **0.272**\*\*
*0.309*\*
0.389
0.555
0.365 | **-0.103**\*\*
*0.147*\*
-0.403
-0.214
1.120 |
| Non-linear Latent (no $U \to X$)
True ATE: 0.850 | -0.316 | -0.423 | TrueIV
ZNet
AutoIV
VIV
GIV | *-0.479*
-0.924
**-0.146**
-0.605
-0.810 | **-0.231**
-0.260
*-0.232*
-0.319
-0.238 | *-0.155*\*
-0.370
0.258
**-0.042**\*\*
-2.706 |
| Non-linear Latent
True ATE: 0.333 | 0.528 | 0.050 | TrueIV
ZNet
AutoIV
VIV
GIV | 1.381
**0.152**
-4.809
1.790
*-0.235* | *-0.020*
-0.039
**-0.008**\*\*
-0.039
-0.028 | 4.762
**-0.063**\*\*
0.785
-0.170
*0.084*\* |
| Non-linear Mixed (no $U \to X$)
True ATE: 1.777 | -0.277 | -0.227 | TrueIV
ZNet
AutoIV
VIV
GIV | **-0.227**\*
-0.443
1.033
*0.438*
15.700 | -0.519
**-0.173**\*
-0.457
*-0.190*\*
-0.334 | -0.723
*-0.196*\*
-0.439
**0.181**\*\*
0.794 |
| Non-linear Mixed
True ATE: 0.558 | 0.849 | 0.255 | TrueIV
ZNet
AutoIV
VIV
GIV | *0.477*
**0.244**\*
10.821
0.950
-0.981 | *0.142*\*
0.218
**0.036**\*\*
0.408
0.293 | *-0.156*\*
**0.033**\*\*
2.079
0.847
0.983 |
| Non-linear No Candidate (no $U \to X$)
True ATE: 0.828 | 0.174 | 0.134 | TrueIV
ZNet
AutoIV
VIV
GIV | –
*0.267*
-0.445
**0.111**
-0.405 | –
*0.117*\*
**0.069**\*\*
0.164
0.291 | –
0.302
*0.275*\*
**0.043**\*\*
-0.277 |
| Non-linear No Candidate (no $U$)
True ATE: 1.429 | 0.250 | -0.068 | TrueIV
ZNet
AutoIV
VIV
GIV | –
*-0.528*
-1.806
**0.182**\*\*
3.389 | –
**-0.012**\*\*
0.064
-0.085
*0.053* | –
**-0.143**\*\*
-0.257
*-0.209*\*
-0.665 |
| Non-linear No Candidate
True ATE: 0.435 | 0.783 | 0.423 | TrueIV
ZNet
AutoIV
VIV
GIV | –
*0.200*\*
-25.181
0.898
**-0.109** | –
**0.260**\*\*
0.720
*0.422*\*
0.640 | –
**0.049**\*\*
0.477
0.404
*0.345*\* |

Table 6: **ATE results on synthetic non-Linear datasets**.

| Dataset | IV Method | F-Stat(Z,T) (Relevance) (Train/Val/Test) | Corr(Z,C) (Independence) (Train/Val/Test) | Corr(Z,Y-Hat) (Exogeneity) (Train/Val/Test) | Corr(Z,U) (Independence) (Train/Val/Test) |
|---|---|---|---|---|---|
| Linear Disjoint | TrueIV | 53.566 / 12.507 / 6.603 | 0.027 / 0.051 / 0.136 | 0.047 / 0.116 / 0.117 | 0.030 / 0.062 / 0.045 |
| | ZNet | 63.617 / 14.699 / 4.927 | 0.040 / 0.044 / 0.092 | 0.037 / 0.059 / 0.060 | 0.140 / 0.175 / 0.088 |
| | AutoIV | 22.820 / 6.721 / 0.944 | 0.214 / 0.211 / 0.253 | 0.000 / 0.000 / 0.000 | 0.166 / 0.143 / 0.272 |
| | VIV | 18.407 / 1.497 / 4.065 | 0.038 / 0.064 / 0.116 | 0.018 / 0.030 / 0.026 | 0.036 / 0.035 / 0.085 |
| | GIV | 1.301 / 0.642 / 6.703 | 0.135 / 0.114 / 0.161 | 0.001 / 0.110 / 0.072 | 0.194 / 0.184 / 0.199 |
| Linear Disjoint (no $U \rightarrow X$) | TrueIV | 35.681 / 29.173 / 4.562 | 0.027 / 0.051 / 0.136 | 0.066 / 0.103 / 0.231 | 0.018 / 0.038 / 0.065 |
| | ZNet | 13.524 / 13.151 / 6.735 | 0.040 / 0.090 / 0.058 | 0.013 / 0.004 / 0.102 | 0.022 / 0.050 / 0.095 |
| | AutoIV | 10.865 / 2.572 / 3.442 | 0.186 / 0.184 / 0.260 | 0.000 / 0.000 / 0.000 | 0.014 / 0.060 / 0.080 |
| | VIV | 18.639 / 6.582 / 3.192 | 0.037 / 0.076 / 0.077 | 0.037 / 0.092 / 0.142 | 0.060 / 0.053 / 0.052 |
| | GIV | 0.278 / 0.253 / 0.788 | 0.148 / 0.161 / 0.195 | 0.001 / 0.065 / 0.060 | 0.011 / 0.041 / 0.068 |
| Linear Latent | TrueIV | 68.564 / 36.180 / 30.735 | 0.200 / 0.201 / 0.250 | 0.027 / 0.007 / 0.046 | 0.018 / 0.053 / 0.085 |
| | ZNet | 210.017 / 29.806 / 13.066 | 0.180 / 0.196 / 0.200 | 0.035 / 0.034 / 0.089 | 0.112 / 0.185 / 0.097 |
| | AutoIV | 47.460 / 21.996 / 22.106 | 0.263 / 0.256 / 0.271 | 0.000 / 0.000 / 0.000 | 0.051 / 0.055 / 0.088 |
| | VIV | 13.772 / 1.628 / 1.948 | 0.031 / 0.060 / 0.118 | 0.022 / 0.017 / 0.109 | 0.022 / 0.050 / 0.102 |
| | GIV | 7.756 / 1.936 / 0.007 | 0.140 / 0.164 / 0.166 | 0.036 / 0.036 / 0.141 | 0.021 / 0.066 / 0.046 |
| Linear Latent (no $U \rightarrow X$) | TrueIV | 68.564 / 36.180 / 30.735 | 0.204 / 0.212 / 0.266 | 0.009 / 0.019 / 0.039 | 0.018 / 0.053 / 0.085 |
| | ZNet | 25.361 / 9.822 / 10.956 | 0.042 / 0.053 / 0.146 | 0.009 / 0.063 / 0.078 | 0.012 / 0.056 / 0.094 |
| | AutoIV | 38.894 / 19.722 / 4.108 | 0.269 / 0.275 / 0.268 | 0.000 / 0.000 / 0.000 | 0.023 / 0.073 / 0.116 |
| | VIV | 23.563 / 13.233 / 5.743 | 0.023 / 0.047 / 0.134 | 0.007 / 0.019 / 0.095 | 0.028 / 0.055 / 0.041 |
| | GIV | 4.305 / 0.010 / 0.621 | 0.137 / 0.141 / 0.137 | 0.003 / 0.057 / 0.065 | 0.021 / 0.039 / 0.072 |
| Linear Mixed | TrueIV | 26.792 / 7.467 / 10.101 | 0.027 / 0.051 / 0.136 | 0.176 / 0.120 / 0.272 | 0.030 / 0.062 / 0.045 |
| | ZNet | 24.163 / 11.648 / 9.514 | 0.168 / 0.166 / 0.233 | 0.016 / 0.025 / 0.101 | 0.059 / 0.071 / 0.115 |
| | AutoIV | 77.162 / 20.885 / 12.485 | 0.236 / 0.243 / 0.287 | 0.000 / 0.001 / 0.001 | 0.322 / 0.330 / 0.281 |
| | VIV | 10.911 / 3.364 / 6.297 | 0.026 / 0.055 / 0.120 | 0.037 / 0.027 / 0.058 | 0.030 / 0.057 / 0.053 |
| | GIV | 9.928 / 3.886 / 3.115 | 0.143 / 0.172 / 0.232 | 0.022 / 0.081 / 0.084 | 0.060 / 0.124 / 0.008 |
| Linear Mixed (no $U \rightarrow X$) | TrueIV | 33.874 / 20.091 / 16.317 | 0.027 / 0.051 / 0.136 | 0.004 / 0.033 / 0.164 | 0.018 / 0.038 / 0.065 |
| | ZNet | 207.114 / 39.401 / 27.046 | 0.096 / 0.092 / 0.153 | 0.010 / 0.101 / 0.081 | 0.014 / 0.049 / 0.090 |
| | AutoIV | 1.394 / 0.111 / 0.225 | 0.261 / 0.272 / 0.253 | 0.000 / 0.000 / 0.000 | 0.022 / 0.040 / 0.116 |
| | VIV | 9.804 / 0.683 / 4.310 | 0.032 / 0.051 / 0.091 | 0.035 / 0.026 / 0.187 | 0.041 / 0.062 / 0.122 |
| | GIV | 9.271 / 0.775 / 0.506 | 0.148 / 0.143 / 0.152 | 0.013 / 0.062 / 0.076 | 0.017 / 0.029 / 0.037 |
| Linear No Candidate | TrueIV | – | – | – | – |
| | ZNet | 21.299 / 2.298 / 2.315 | 0.037 / 0.060 / 0.122 | 0.010 / 0.089 / 0.050 | 0.223 / 0.228 / 0.247 |
| | AutoIV | 77.352 / 30.622 / 0.391 | 0.265 / 0.263 / 0.241 | 0.000 / 0.000 / 0.000 | 0.368 / 0.357 / 0.343 |
| | VIV | 18.753 / 6.859 / 3.228 | 0.042 / 0.063 / 0.095 | 0.060 / 0.056 / 0.096 | 0.065 / 0.076 / 0.090 |
| | GIV | 3.667 / 1.458 / 0.825 | 0.138 / 0.158 / 0.166 | 0.009 / 0.027 / 0.012 | 0.126 / 0.150 / 0.216 |
| Linear No Candidate (no $U \rightarrow X$) | TrueIV | – | – | – | – |
| | ZNet | 78.441 / 1.012 / 7.704 | 0.169 / 0.168 / 0.214 | 0.008 / 0.093 / 0.023 | 0.056 / 0.061 / 0.088 |
| | AutoIV | 25.720 / 0.402 / 0.701 | 0.254 / 0.268 / 0.264 | 0.000 / 0.000 / 0.000 | 0.012 / 0.053 / 0.051 |
| | VIV | 26.796 / 6.537 / 16.852 | 0.029 / 0.070 / 0.134 | 0.038 / 0.068 / 0.075 | 0.048 / 0.069 / 0.071 |
| | GIV | 0.298 / 0.272 / 0.029 | 0.200 / 0.193 / 0.197 | 0.004 / 0.020 / 0.008 | 0.017 / 0.057 / 0.057 |
| Linear No Candidate (no $U$) | TrueIV | – | – | – | – |
| | ZNet | 463.273 / 3.552 / 0.938 | 0.110 / 0.118 / 0.095 | 0.074 / 0.015 / 0.083 | – / – / – |
| | AutoIV | 18.562 / 0.437 / 3.744 | 0.248 / 0.252 / 0.206 | 0.000 / 0.000 / 0.000 | – / – / – |
| | VIV | 15.274 / 7.716 / 4.874 | 0.030 / 0.062 / 0.103 | 0.021 / 0.108 / 0.144 | – / – / – |
| | GIV | 0.055 / 2.459 / 2.216 | 0.156 / 0.143 / 0.208 | 0.022 / 0.018 / 0.041 | – / – / – |

Table 7: **Instrument strength and validity on linear synthetic datasets**.

| Dataset | IV Method | F-Stat(Z,T) (Relevance) (Train/Val/Test) | Corr(Z,C) (Independence) (Train/Val/Test) | Corr(Z,Y-Yhat) (Exogeneity) (Train/Val/Test) | Corr(Z,U) (Independence) (Train/Val/Test) |
|---|---|---|---|---|---|
| Non-linear Disjoint | TrueIV | 41.662 / 12.868 / 4.620 | 0.027 / 0.051 / 0.136 | 0.124 / 0.100 / 0.176 | 0.030 / 0.062 / 0.045 |
| | ZNet | 28.150 / 12.159 / 19.772 | 0.067 / 0.077 / 0.111 | 0.004 / 0.028 / 0.107 | 0.093 / 0.131 / 0.069 |
| | AutoIV | 142.891 / 48.571 / 15.159 | 0.214 / 0.232 / 0.205 | 0.000 / 0.000 / 0.000 | 0.327 / 0.287 / 0.357 |
| | VIV | 16.443 / 1.351 / 10.179 | 0.038 / 0.054 / 0.111 | 0.015 / 0.079 / 0.015 | 0.030 / 0.068 / 0.073 |
| | GIV | 37.947 / 8.967 / 17.319 | 0.145 / 0.120 / 0.184 | 0.021 / 0.027 / 0.045 | 0.187 / 0.187 / 0.266 |
| Non-linear Disjoint (no $U \to X$) | TrueIV | 19.956 / 5.440 / 18.102 | 0.027 / 0.051 / 0.136 | 0.014 / 0.014 / 0.133 | 0.018 / 0.038 / 0.065 |
| | ZNet | 17.203 / 4.722 / 12.113 | 0.038 / 0.058 / 0.170 | 0.013 / 0.065 / 0.075 | 0.018 / 0.045 / 0.071 |
| | AutoIV | 5.259 / 3.103 / 1.239 | 0.225 / 0.224 / 0.233 | 0.000 / 0.000 / 0.000 | 0.033 / 0.016 / 0.084 |
| | VIV | 4.690 / 0.937 / 1.670 | 0.035 / 0.069 / 0.098 | 0.004 / 0.070 / 0.128 | 0.044 / 0.056 / 0.070 |
| | GIV | 16.805 / 1.806 / 5.565 | 0.130 / 0.133 / 0.237 | 0.114 / 0.006 / 0.140 | 0.035 / 0.071 / 0.036 |
| Non-linear Latent | TrueIV | 30.818 / 7.298 / 2.185 | 0.204 / 0.212 / 0.266 | 0.005 / 0.011 / 0.052 | 0.064 / 0.083 / 0.022 |
| | ZNet | 19.834 / 1.402 / 1.290 | 0.034 / 0.043 / 0.172 | 0.004 / 0.014 / 0.070 | 0.116 / 0.089 / 0.081 |
| | AutoIV | 76.120 / 14.009 / 0.317 | 0.212 / 0.192 / 0.205 | 0.000 / 0.000 / 0.000 | 0.363 / 0.328 / 0.311 |
| | VIV | 16.454 / 4.075 / 3.042 | 0.027 / 0.052 / 0.118 | 0.018 / 0.061 / 0.024 | 0.051 / 0.064 / 0.132 |
| | GIV | 20.388 / 4.855 / 11.145 | 0.123 / 0.130 / 0.145 | 0.001 / 0.082 / 0.239 | 0.226 / 0.266 / 0.328 |
| Non-linear Latent (no $U \to X$) | TrueIV | 35.494 / 1.009 / 0.196 | 0.204 / 0.212 / 0.266 | 0.002 / 0.030 / 0.050 | 0.018 / 0.053 / 0.085 |
| | ZNet | 19.330 / 0.650 / 0.817 | 0.028 / 0.069 / 0.109 | 0.012 / 0.038 / 0.139 | 0.016 / 0.046 / 0.086 |
| | AutoIV | 43.267 / 1.545 / 0.663 | 0.239 / 0.257 / 0.234 | 0.000 / 0.000 / 0.000 | 0.044 / 0.056 / 0.104 |
| | VIV | 10.032 / 8.944 / 12.677 | 0.025 / 0.056 / 0.133 | 0.034 / 0.094 / 0.142 | 0.047 / 0.074 / 0.085 |
| | GIV | 0.302 / 0.010 / 2.774 | 0.100 / 0.100 / 0.115 | 0.023 / 0.025 / 0.107 | 0.037 / 0.056 / 0.064 |
| Non-linear Mixed | TrueIV | 39.277 / 14.385 / 6.638 | 0.027 / 0.051 / 0.136 | 0.080 / 0.084 / 0.186 | 0.030 / 0.062 / 0.045 |
| | ZNet | 81.609 / 27.086 / 14.220 | 0.037 / 0.038 / 0.140 | 0.013 / 0.024 / 0.088 | 0.099 / 0.084 / 0.062 |
| | AutoIV | 218.811 / 58.758 / 33.881 | 0.201 / 0.222 / 0.227 | 0.000 / 0.000 / 0.000 | 0.345 / 0.301 / 0.358 |
| | VIV | 13.916 / 7.281 / 4.760 | 0.021 / 0.070 / 0.132 | 0.021 / 0.033 / 0.154 | 0.038 / 0.060 / 0.078 |
| | GIV | 8.884 / 2.184 / 4.662 | 0.096 / 0.142 / 0.184 | 0.055 / 0.098 / 0.017 | 0.045 / 0.097 / 0.055 |
| Non-linear Mixed (no $U \to X$) | TrueIV | 60.302 / 11.847 / 7.502 | 0.027 / 0.051 / 0.136 | 0.034 / 0.068 / 0.144 | 0.018 / 0.038 / 0.065 |
| | ZNet | 972.072 / 27.721 / 14.899 | 0.038 / 0.059 / 0.100 | 0.022 / 0.036 / 0.044 | 0.038 / 0.029 / 0.032 |
| | AutoIV | 354.559 / 71.052 / 57.682 | 0.270 / 0.274 / 0.322 | 0.000 / 0.000 / 0.000 | 0.026 / 0.051 / 0.082 |
| | VIV | 16.785 / 3.347 / 1.459 | 0.038 / 0.049 / 0.113 | 0.021 / 0.070 / 0.204 | 0.048 / 0.073 / 0.081 |
| | GIV | 3.221 / 5.328 / 0.129 | 0.153 / 0.171 / 0.180 | 0.116 / 0.010 / 0.137 | 0.016 / 0.052 / 0.079 |
| Non-linear No Candidate | TrueIV | – | – | – | – |
| | ZNet | 15.335 / 4.959 / 1.181 | 0.067 / 0.102 / 0.222 | 0.020 / 0.048 / 0.093 | 0.119 / 0.099 / 0.127 |
| | AutoIV | 85.371 / 16.582 / 0.149 | 0.227 / 0.211 / 0.302 | 0.000 / 0.000 / 0.000 | 0.205 / 0.165 / 0.243 |
| | VIV | 18.972 / 3.681 / 5.362 | 0.025 / 0.054 / 0.113 | 0.027 / 0.084 / 0.089 | 0.022 / 0.060 / 0.100 |
| | GIV | 16.638 / 4.238 / 4.860 | 0.173 / 0.155 / 0.165 | 0.059 / 0.031 / 0.010 | 0.132 / 0.051 / 0.086 |
| Non-linear No Candidate (no $U \to X$) | TrueIV | – | – | – | – |
| | ZNet | 102.430 / 2.654 / 3.266 | 0.046 / 0.071 / 0.138 | 0.010 / 0.056 / 0.057 | 0.067 / 0.041 / 0.049 |
| | AutoIV | 99.004 / 21.237 / 14.164 | 0.192 / 0.186 / 0.252 | 0.000 / 0.000 / 0.000 | 0.022 / 0.089 / 0.048 |
| | VIV | 13.953 / 18.708 / 2.071 | 0.033 / 0.082 / 0.116 | 0.043 / 0.035 / 0.100 | 0.028 / 0.085 / 0.089 |
| | GIV | 2.033 / 4.684 / 1.203 | 0.183 / 0.169 / 0.198 | 0.015 / 0.009 / 0.042 | 0.025 / 0.024 / 0.080 |
| Non-linear No Candidate (no $U$) | TrueIV | – | – | – | – |
| | ZNet | 123.260 / 10.146 / 2.912 | 0.208 / 0.223 / 0.216 | 0.117 / 0.107 / 0.166 | – / – / – |
| | AutoIV | 75.873 / 14.926 / 8.766 | 0.223 / 0.201 / 0.268 | 0.000 / 0.000 / 0.000 | – / – / – |
| | VIV | 16.756 / 8.589 / 9.893 | 0.032 / 0.062 / 0.114 | 0.068 / 0.078 / 0.079 | – / – / – |
| | GIV | 10.804 / 5.268 / 0.193 | 0.137 / 0.120 / 0.194 | 0.000 / 0.056 / 0.038 | – / – / – |

Table 8: **Instrument strength and validity on non-linear synthetic datasets**.

| | TSLS Avg. \|Error ATE\| (SE) | DeepIV Avg. \|Error ATE\| (SE) | DFIV Avg. \|Error ATE\| (SE) |
|---|---|---|---|
| ZNet | **0.550 (0.154)** | **0.201 (0.040)** | **0.271 (0.042)** |
| AutoIV | 3.305 (1.416) | 0.262 (0.050) | 0.675 (0.142) |
| VIV | 0.776 (0.222) | 0.254 (0.042) | 0.316 (0.056) |
| GIV | 3.192 (1.574) | 0.285 (0.039) | 0.891 (0.319) |

Table 9: **Comparison of IV methods on average across the 18 different data generation processes**.

|        | **DeepIV** Avg. PEHE (SE) | **DFIV** Avg. PEHE (SE) |
|--------|---------------------------|-------------------------|
| ZNet   | **0.470 (0.063)**         | *0.552 (0.111)*         |
| AutoIV | 0.559 (0.060)             | 0.882 (0.147)           |
| VIV    | 0.586 (0.059)             | **0.491 (0.056)**       |
| GIV    | 0.519 (0.057)             | 0.999 (0.3106)          |

Table 10: **Comparison of IV methods on average across the 18 different data generation processes for CATE estimation**.

