# OpenReview forum: "Causal Effect Estimation with Learned Instrument Representations"
_ICLR.cc/2026/Conference — Submitted to ICLR 2026_

### Official Review · Reviewer_3W6s · 2025-10-29

**Soundness:** 2
**Presentation:** 2
**Contribution:** 2
**Rating:** 4
**Confidence:** 4

**Summary:**

This paper introduces ZNet, a novel deep learning framework that addresses the critical challenge of unobserved confounding in causal effect estimation from observational data. Its primary contribution lies in automatically constructing valid instrumental variables (IVs) by decomposing observed variables into latent representations that satisfy core IV assumptions—relevance, exclusion restriction, and unconfoundedness. Unlike existing methods that require pre-specified candidate IVs, ZNet's architecture explicitly encodes the structural causal model of IVs, enabling it to function as a plug-in module for various two-stage IV estimators. Experimental results demonstrate that ZNet can both recover ground-truth instruments when available and generate proxy latent instruments that effectively reduce estimation bias when no explicit IVs exist, offering a flexible solution for causal inference in general observational settings where unconfoundedness cannot be verified.

**Strengths:**

The paper demonstrates significant originality by shifting the paradigm from selecting candidate instruments to constructing them directly from observed data. Unlike methods that require pre-specified IV candidates, ZNet automatically decomposes covariates into valid instrumental and confounder representations through a structurally-informed architecture, creatively combining deep learning with causal constraints.

This conceptual contribution is well-supported by high methodological rigor. ZNet's losses directly enforce IV assumptions, and comprehensive experiments show it both recovers true instruments and constructs effective proxies when none exist. Its robust performance across diverse settings—including when unobserved confounding is absent—underscores its practical significance as a reliable, plug-in causal estimator.

**Weaknesses:**

While the paper positions ZNet as a novel paradigm for IV construction, the core idea of learning instrumental variable representations from data is an established research direction.  Several prior works have the explicit goal of generating valid IVs from observed covariates without pre-specified candidates, using variational and mutual information-based approaches. The architectural difference of ZNet—learning a decomposition based on an SCM—is a technical contribution but does not constitute a paradigm shift.

A more critical weakness is the exclusive reliance on semi-synthetic datasets for experimental validation.   While useful for controlled testing with known ground truth, this setup fails to demonstrate the method's applicability to real-world problems. Furthermore, the absence of an evaluation on a real-world dataset with no ground-truth effect (e.g., a standard observational causal inference benchmark) leaves the practical utility in doubt.  For the method to be considered a "plug-in" solution, it must be validated in settings that mirror the true uncertainty of observational studies.

"Moreover, all code to generate models, synthetic data, and experiments will be made public upon publication." I suggest publish in the review period.

**Questions:**

Please see my concerns in weakness.

---

> ### Author Response · Authors · 2025-11-25
>
> We thank the reviewer for recognizing the novelty of our contributions and the breadth of our evaluation. We also appreciate the constructive feedback and address each of the concerns raised in the weaknesses below.
>
> **Weakness 1:** We would like to emphasize our contribution to the literature beyond existing works, which we add language to highlight in Section 7 of the revision.
>
> **Discriminatory approach to IV generation:** First, we present a novel architecture, which is not only discriminatory rather than generative but also higher performing on average in ATE estimation across various data generation processes (see the table below). Existing methods use VAEs to learn data distributions that enforce independence conditions. As you point out, we learn functions that resemble structural equations, providing a lightweight, straightforward and interpretable discriminatory method of generating instrument representations.
>
> **We do not assume unconfoundedness of observed data:** Second, we provide a method that makes fewer assumptions than existing literature. To our knowledge, all existing methods assume that the observed confounders are not influenced by the unobserved confounders in order to generate an instrument representation that will not be confounded. We use lemma 1 to construct a loss which removes confounding by any unobserved confounders influencing the observed data from the generated representation.
>
> **Comprehensive evaluation of IV generation:** Finally, we contribute the most extensive evaluation of IV generation. We show that ZNet can recover a suitable instrument representation by comparing its relevance, exclusion restriction, and unconfoundedness empirically across 18 datasets with data generation processes that span a wide variety of real-world scenarios (Appendix Table 7, 8). The generated IVs are suitable for ATE (Table 1, Appendix Tables 5, 6) and CATE estimation (Appendix Tables 3, 4) and highest performing on average (see table below). Moreover, the generated IV is highly correlated with any existing instruments, both observed (Figure 5, Appendix Figure 6) and latent (Figure 4). This analysis across settings suggests that IV generation is broadly applicable, which is a novel contribution in and of itself, i.e. the extensive evaluation suggests that ZNet’s IV generation can be used as a plug in causal inference method.
>
>
> **Weakness 2:** Semi-synthetic datasets are standard in evaluating causal inference methods since we can evaluate the true recovery of causal effects with precision (see the combined response above for additional details). Literature in causal inference evaluates on real-world world data for which the treatment effects and outcomes are manipulated in order to be measurable. We do the same. Our datasets use the standard causal inference benchmark IHDP (Hill 2011). Adding real world data, where true treatment effects cannot be known, would not serve to compare the performance of ZNet with existing causal inference models.
>
> **Weakness 3:** We are excited to make the codebase available for use. However, we do not see the value of making this available during the rebuttal period unless the reviewer provides more context on the aims in assessing the codebase. We instead plan to finish the proper documentation necessary to make this resource both available and easy to use along the timeline of paper publication.
>
> Jennifer L Hill. Bayesian nonparametric modeling for causal inference. Journal of Computational
> and Graphical Statistics, 20(1):217–240, 2011

---

### Official Review · Reviewer_81B1 · 2025-10-30

**Soundness:** 2
**Presentation:** 2
**Contribution:** 2
**Rating:** 2
**Confidence:** 5

**Summary:**

This paper addresses the practical challenge of identifying valid instrumental variables (IVs) by proposing ZNet, a decomposition-based framework that learns latent representations satisfying the standard IV assumptions. When real instrumental variables exist in the data, ZNet is able to recover them; when no explicit IVs are available, ZNet can construct proxy latent IVs to facilitate causal effect estimation.

**Strengths:**

1.The proposed method is conceptually simple and easy to understand.

2.The experiments are comprehensive and cover diverse data scenarios.

**Weaknesses:**

1.Lack of clear novelty. Recovering instrumental variables from observational data is a well-known open problem, and several prior works (e.g., DIV.VAE, GDIV, AutoIV) have explored similar directions. The authors should clarify the key distinctions between ZNet and these existing approaches in terms of motivation, design, or theoretical guarantees.

2. Equations 5 and 6 fail to demonstrate the lack of confounding in instrumental variables. In Equation 5, the outcome variable Y is directly predicted using the covariate X and the treatment variable T, which inherently incorporates confounding factors. Furthermore, in Equation 6, the meaning of Y - \hat_{Y} is unclear. Why does its uncorrelated nature with Z guarantee that the instrumental variable Z is free from confounding?

3. Similarly, exclusivity and relevance cannot be satisfied. Crucially, how does the author ensure that f(X) and g(X) capture the confounding factor C and the instrumental variable Z, respectively? Without clarifying this key point, the validity of Equations 7–9 cannot be guaranteed.

4. The superiority of ZNet cannot be assured. Based on the experimental data in Table 1, ZNet does not demonstrate outstanding performance.

5. The definition of an IV ("An IV is a variable that influences the treatment but has no direct effect on the out come or influence from
 unobserved confounders.") in abstact is wrong.

6. The second condition, i.e., "Exclusion restriction" (page 2) is wrong. It should be Z⊥Y|(T,C, e_Y).

7.  CATE(C)=E[Y|do(T)=1,C]−E[Y|do(T)=0,C], and  ATE=E_C[CATE] seems to be wrong. Based on the standard definition of ATE,
ATE should be E[Y|do(T)=1]−E[Y|do(T)=0], rather than CATE(C).

**Questions:**

See Weaknesses.

---

> ### Author Response · Authors · 2025-11-25
>
> We thank the reviewer for careful review and thoughtful constructive feedback. The feedback highlights several places where the exposition of our contribution and its novelty can be improved, which we appreciate. We address all concerns point by point below and include clarifications in the revision.
>
> **Weakness 1:** We appreciate that there are a handful of works already exploring IV generation. We compare the motivation, assumptions, and design choices made by these works to our own in the table below. We would like to clarify that our contribution is novel for three key reasons which we further emphasize in Section 7 of the revision.
>
> **Discriminatory approach to IV generation:** First, we present a novel architecture, which is higher performing on average for ATE estimation across various data generation processes (see the table below in reply to Weakness 4). Existing methods use VAEs to learn data distributions that enforce independence conditions. We learn functions that resemble structural equations, providing a lightweight, straightforward and interpretable discriminatory method of generating instrument representations.
>
> **We do not assume unconfoundedness of observed data:** Second, we provide a method that makes fewer assumptions than existing literature. To our knowledge, all existing methods assume that the observed confounders are not influenced by the unobserved confounders in order to generate an instrument representation that will not be confounded. We use lemma 1 to construct a loss which removes confounding by any unobserved confounders influencing the observed data from the generated representation.
>
> **Comprehensive evaluation of IV generation:** Finally, we contribute the most extensive evaluation of IV generation. We show that ZNet can recover a suitable instrument representation by comparing its relevance, exclusion restriction, and unconfoundedness empirically across 18 datasets with data generation processes that span a wide variety of real-world scenarios (Appendix Table 7, 8). The generated IVs are suitable for ATE (Table 1, Appendix Tables 5, 6) and CATE estimation (Appendix Tables 3, 4) and highest performing on average (see table below comparing main text Table 1 and new appendix tables added in revision comparing all data). Moreover, ZNet generated IVs can be highly correlated with any existing instruments, both observed (Figure 5, Appendix Figure 6) and latent (Figure 4). This analysis across settings suggests that IV generation is broadly applicable, which is a novel contribution in and of itself, i.e. the extensive evaluation suggests that ZNet’s IV generation can be used as a plug in causal inference method.
>
> | Method   | Motivation                                                                 | Assumes IVs exist among X | Assumes latent IVs influence X | Assumes X is unconfounded | Discriminatory approach | Generative approach |
> |----------|---------------------------------------------------------------------------|---------------------------|--------------------------------|--------------------------|-----------------------|-------------------|
> | DIV.VAE  | Deconstruct observed data to recover surrogate IV                          | Yes (by proxy)            | No                             | Yes                      | No                    | Yes               |
> | DVAE.CIV | Deconstruct observed data to recover conditional IV                        | Yes                       | No                             | Yes                      | No                    | Yes               |
> | GDIV    | Recover IV from graphical data                                            | No                        | No                             | Yes                      | No                    | Yes               |
> | GIV      | Recover latent environment IV to disambiguate datasets                     | No                        | Yes                            | Yes                      | No                    | Yes               |
> | AutoIV   | Learn a latent decomposition of the observed data and generate an IV       | No                        | No                             | Yes                      | No                    | Yes               |
> | VIV      | Learn a latent decomposition of the observed data and generate an IV       | No                        | No                             | Yes                      | No                    | Yes               |
> | ZNet     | Discriminate an IV representation that generates or recovers an IV        | No                        | No                             | No                       | Yes                   | No                |

---

> ### Author Response · Authors · 2025-11-25
>
> **Weakness 2:** Thank you for the constructive feedback that the presentation of the loss construction is not clear. We appreciate the opportunity to clarify how (5) and (6) guarantee the lack of confounding of IVs in the setting of Lemma 1 and update the paper to better exposit this contribution. The variable Y - \hat{Y} is the residuals of a model predicting Y from the observed data X and treatment T. This is modeled by the difference between a separate network estimating E[Y|X,T] and the observed Y. This model error Y - \hat{Y} captures information about the variability of Y due to U. In particular, Y - \hat{Y} = e_Y - E[e_Y |X,T] as is written out in lines 190-191. Lemma 1 guarantees that if Z is normally distributed with mean 0, then zero covariance of Z with  e_Y - E[e_Y |X,T] implies zero covariance of Z with e_Y. The latter is the desired unconfoundedness property. Thus, equations (5) and (6) directly correspond to Lemma 1. We improve the description of this in Sections 3 and 5.1 in revision.
>
> **Weakness 3:** We also appreciate the confusion over the second and third loss constraints and the opportunity to clarify. Exclusion restriction is enforced by making f(X) predictive of Y and f(X) and g(X) independent of each other. This makes g(X) unable to enter the SCM for Y except through T. Relevance is enforced by making g(X) predictive of T. We improve the clarity of this description in the paper in Section 5.1.
>
> We also clarify that f(X) may not explicitly recover the observed data and g(X) may not recover an existing instrument. However, equation (7) guarantees that f(X) recovers information predictive of Y from X and equation (9) guarantees that g(X) recovers information predictive of T. Moreover, the ability of f(X) and g(X) to recover the treatment effects across our evaluations demonstrates their recovery of the information necessary to predict outcomes, which is sufficient.
>
> **Weakness 4:** On average, as shown in the table below summarizing Table 1, ZNet is higher performing at ATE recovery than the other IV generation methods across settings. We intentionally validate under a multitude of data generation processes since the generation process of real data cannot be known. ZNet has high performance across settings compared to the baselines which suggests it is a more generally higher performing IV generation method.
>
> | Method | TSLS      | DeepIV    | DF IV     |
> |--------|---------------|---------------|---------------|
> | ZNet   | **0.507 (0.252)** | **0.163 (0.041)** | **0.212 (0.059)** |
> | AutoIV | 4.863 (2.469) | 0.238 (0.080) | 0.637 (0.177) |
> | VIV    | 0.654 (0.183) | 0.274 (0.067) | 0.351 (0.079) |
> | GIV    | 1.024 (0.353) | 0.263 (0.064) | 0.494 (0.117) |
>
> *Rebuttal Table.* The mean absolute value of the error on ATE across datasets in Table 1 (with standard errors across these 10 datasets) by downstream estimator and IV generation method.

---

> > ### Author Response · Authors · 2025-11-25
> >
> > **Weakness 5:** Thank you for pointing this out. We are happy to revise the sentence “An IV is a variable that influences the treatment but has no direct effect on the outcome or influence from unobserved confounders.” to be more precise as “An IV is a variable that influences the treatment but has no direct effect on the outcome and is independent of unobserved confounders.”
> >
> > **Weakness 6:** Thank you for highlighting this discrepancy. Exclusion restriction requires that Z only enters the structural causal model for Y, i.e. equation (1), through T. In Hartford 2017, using independence conditions, the authors phrase as Z \ind Y | T, C, e_Y. Some other works adopt this notation. Since conditioning on T, C, and e_Y when Y = phi(T, C) + e_Y makes the statement trivially true, we prefer the definition that Z \ind Y | T, C in the causal graph with T \to Y removed. However, as we omitted the clause that T \to Y be removed and this notation is confusing without the introduction of DAGs, we propose changing the phrasing to “Z only enters phi through T”, which has been updated in revision.
> >
> > **Weakness 7:** We appreciate the apparent confusion in our notation. The ATE is the treatment effect averaged (or taken in expectation over) the distribution of all confounders C. We use the law of iterated expectation (total expectation) to write the definition of CATE. However, while accurate, this language may be opaque, and we are happy to change the notation in the revised version.

---

> > > ### Comment · Reviewer_81B1 · 2025-11-28
> > > **Thank you for your responses and revisions.**
> > >
> > > I have read the rebuttal and the revised version.
> > >
> > > The condition for the instrumental variable, specifically “Exclusion restriction: Z only enters \phi through T”, remains incorrect in the revised manuscript. Please restate this assumption following the standard definition of instrumental variables as presented in Pearl (2009, Causality) or in the DeepIV framework.
> > >
> > > In addition, the definition of the CATE requires conditioning on the full set of parents of Y. Conditioning only on the confounders C is insufficient for a correct definition.

---

> ### Author Response · Authors · 2025-12-02
>
> We thank the reviewer for their attention to our revision and add the following clarifications:
>
> **Remark 1:** First, there are several equivalent ways of phrasing the conditions for an instrumental variable. For clarity, we explain how our definition of exclusion restriction fits within these equivalent definitions.
>
> In Pearl 2009, exclusion restriction and unfoundedness requirements are combined into the statement that the IV is “independent of all other variables that have influence on Y not mediated by T.” Pearl notes that this is equivalent to the graphical condition that the IV and Y are independent in the DAG with all arrows into T removed. Exclusion restriction is simply that the IV does not have direct influence on the outcome Y except through T (frequently cited as in Angrist, Imbens, and Rubin 1996 under the Rubin/potential outcomes model as Y (Z, T) = Y (Z’, T) for all T and IV Z). Unconfoundedness is what remains: that the IV be independent of all other confounders for Y, unless we can observe and condition on them.
>
> In Hartford et al 2017 (DeepIV), exclusion restriction and unconfoundedness are listed separately. Exclusion restriction is listed as the condition that the IV does not enter the structural equation for Y. Implicitly, the authors mean that the IV does not enter the structural equation for Y except that effect mitigated through T since the IV must be relevant. This is the definition we choose for its simplicity and alignment with DeepIV. Hartford et al goes on to state this definition equivalently as Z is independent of Y given the treatment, observed data, and unobserved error. We prefer to avoid this confusing notation and keep the former definition.
>
> **Remark 2:** We clarify the misunderstanding that the definition of CATE requires conditioning on the full set of parents of Y. Indeed the challenge in estimating causal effects in the setting of unobserved confounding is that we cannot condition on the unobserved parents of Y. In estimating CATE, we always seek to estimate the treatment effect in a population with the same set of observed data C. The output of any causal effect estimator given inputs T, C, and outputs Y will be this CATE. The observed individual level treatment effect (ITE) will be influenced by U and C. PEHE captures the error in estimating CATE with a given model at predicting the ITE.
>
> Angrist, Imbens, and Rubin (1996). Identification of Causal Effects Using Instrumental Variables. Journal of the American Statistical AssociationJune 1996, Vol. 91, No. 434
>
> Pearl, J. (2009). Causality: Models, Reasoning, and Inference (2nd ed.). Cambridge University Press

---

### Official Review · Reviewer_D82i · 2025-10-31

**Soundness:** 2
**Presentation:** 2
**Contribution:** 2
**Rating:** 6
**Confidence:** 1

**Summary:**

this paper proposes a method for decomposing the observed variables to find a representation which satisfies the standard IV assumptions of relevance, exclusion restriction, and unconfoundedness. The experiments validates the effectiveness of ZNet.

**Strengths:**

Generally clear and readable; figures and tables are informative;

**Weaknesses:**

1. Results are synthetic; real-world case studiesare absent.
2. In some regimes/methods ZNet is not best. A deeper error analysis would help users decide when ZNet is reliable

**Questions:**

N/A

---

> ### Author Response · Authors · 2025-11-25
>
> Thank you for the review of our paper. We respond point by point to the weaknesses.
>
> **Weakness 1:** Semi-synthetic datasets are standard in evaluating causal inference methods since we can evaluate the true recovery of causal effects with precision (see the combined response above for additional details). Literature in causal inference evaluates on real-world world data for which the treatment effects and outcomes are manipulated in order to be measurable. We do the same. Our datasets use the standard causal inference benchmark IHDP (Hill 2011). Adding real world data, where true treatment effects cannot be known, would not serve to compare the performance of ZNet with existing causal inference models.
>
> **Weakness 2:** ZNet performs most reliably and on average the best across all simulated settings (see table below). This is critical because, in real-word settings, we do not know the data generation process. Based on our comprehensive semi-synthetic experiments, choosing ZNet is the ‘safest’ or best when your setting is unknown. Our semi-synthetic experiments are comprehensive. We generate data across settings that span instrument availability (lines 323 to 327), include and exclude potential influence of unobserved confounders on the observed data used to generate the instrument, and include and exclude unobserved confounding (see Table 1, Appendix Tables 5, 6, and novel Appendix Figure 8). We would be excited to consider additional analyses of other data generation processes to observe ZNet’s errors on these compared to other data should the reviewer have suggestions.
>
> | Method | TSLS      | DeepIV    | DF IV     |
> |--------|---------------|---------------|---------------|
> | ZNet   | **0.507 (0.252)** | **0.163 (0.041)** | **0.212 (0.059)** |
> | AutoIV | 4.863 (2.469) | 0.238 (0.080) | 0.637 (0.177) |
> | VIV    | 0.654 (0.183) | 0.274 (0.067) | 0.351 (0.079) |
> | GIV    | 1.024 (0.353) | 0.263 (0.064) | 0.494 (0.117) |
>
> *Rebuttal Table.* The mean absolute value of the error on ATE across datasets in Table 1 (with standard errors across these 10 datasets) by downstream estimator and IV generation method. ZNet is the superior method on average across comprehensive data generation processes.
>
> Jennifer L Hill. Bayesian nonparametric modeling for causal inference. Journal of Computational
> and Graphical Statistics, 20(1):217–240, 2011

---

### Official Review · Reviewer_Y1uN · 2025-11-05

**Soundness:** 2
**Presentation:** 2
**Contribution:** 2
**Rating:** 2
**Confidence:** 4

**Summary:**

The paper proposes a novel method to learn instrumental variables from the confounders in the dataset. With the help of the learned instruments, treatment effects can be identified and estimated even in the presence of unobserved confounders. Based on an additive loss including multiple single-task losses, the final estimate is forced to comply with the necessary assumptions of IVs. The paper empirically validates the new method ZNet.

**Strengths:**

- The paper includes a rather extensive empirical evaluation.
- To the best of my knowledge, the idea of learning instruments (not instrument representations) from the existing dataset, fulfilling the necessary requirements for valid IVs, is novel.

**Weaknesses:**

- The paper does not include any mathematical guarantees for the validity of the method. Therefore, it cannot be guaranteed that the learned IVs are actually valid IVs, potentially rendering the causal effect unidentifiable. I consider this the main and very severe weakness of the work.
- The paper assumes that there exists a subset of the confounders that can serve as instruments. However, this is not necessarily the case in practice.
- The paper makes the strong assumption that the instrument is sampled from a normal distribution. The reasoning behind this assumption is neither explained nor is the validity discussed.
- The provided loss is only suitable for valid IVs if the final loss terms related to independence requirements are truly 0. However, this is neither shown nor discussed (neither theoretically nor empirically).
- Evaluation: The method is only evaluated for ATE estimation based on the mean error. For better (and fairer) comparison, the method should also be evaluated for CATE estimation based on the PEHE.

**Questions:**

- Very likely, the confounders will not include suitable instruments. How does the method (theoretically) behave in this failure mode? In this case, the treatment effect is theoretically still not point-identified.
- Lemma 1: With which justification can Z be assumed to stem from a normal distribution? How would one define the variance of the distribution?
- Lines 235-237: What is the reasoning behind this model specification?
- Why is the PC-loss a good choice for the loss term? Why not use the HSIC? This is not discussed. Furthermore, to evaluate the loss, the covariance and standard deviations need to be estimated. What happens if the estimation is incorrect?
- How does the method perform for continuous treatments? Here, the representations might be more difficult to learn.

---

> ### Author Response · Authors · 2025-11-25
>
> We thank the reviewer for their detailed and constructive feedback. First, we want to clarify that our method does learn instrument representations. **We do not assume there are instruments available in the confounders but instead learn a representation that serves as an instrument.** In the case that there exist true instruments among the confounders, our method learns instrument representations with high correlation with the original instruments. This is shown empirically true in data regardless of whether original instruments are among the observed confounders or latent variables influencing observed confounders (Figures 4, 5). While recovering the original instrument is exciting, we emphasize that a learned ZNet instrument representation can still be valid even if uncorrelated with the original instrument, as long as the representation satisfies all IV constraints. We change language to further clarify this broader contribution in Sections 1 and 3 based on this confusion (see revised version). Below we provide a point by point response to your additional concerns and questions.
>
> **Weakness 1:** Solutions to the ZNet loss minimization problem will always give a representation that serves as an instrument since IV constraints are explicitly embedded in the loss function. This instrument can then be used in any downstream instrument regression where satisfying the standard IV criteria (or, equivalently, ZNet criteria) implies the validity of subsequent causal inference. Note that the resulting treatment effect estimate of ZNet has the same causal effect guarantees of downstream regression methods (i.e. Newey and Powell 2003). Beyond this, theoretical guarantees on the validity of the ZNet method at generating the best suitable amongst all suitable instruments would be asymptotic results which do not suggest practical considerations for learning. Thus, we focus on the practical and empirical evaluation of learned instruments across a variety of controlled settings. In response to your feedback, we added discussion of this to the discussion in revision.
>
> **Weakness 2/Question 1:** We appreciate the opportunity to clarify that we do not assume the existence of a subset of the confounders serves as an instrument and update language to clarify this in the revision of the paper in sections 1 and 3.
>
> **Weakness 3/Question 2:** We also appreciate the opportunity to clarify that we do not assume that instruments come from a normal distribution. We generate instrument representations that are approximately normally distributed. We do this using a KL divergence loss between a normal distribution and the generated instrument representation. The point of this condition is for regularization and to allow for the satisfaction of the loss terms in (5), (6) to guarantee unconfoundedness (via Lemma 1). In the case that an existing instrument in the confounders is not normally distributed, our instrument representation can still be normal. It may be highly correlated with the true instrument as we observe experimentally (Section 6.2, Figure 4,5) or an abstract representation. We provide language to clarify this in the paper in section 3 (lines 199-200).
>
> **Weakness 4:** We clarify that we do empirically validate the suitability of the instruments generated by our loss terms in Appendix Tables 7, 8. Instrument candidates in real datasets are typically evaluated for their relevance via their F-statistic and assessed with domain knowledge to prevent prohibited relationships. We perform evaluations on the ZNet generated instrument using the F-statistic and through the correlation of the instrument with the confounders. These analyses are referenced in the manuscript in line 381. In response to this comment, we add a main text figure depicting a sample of these results. Does the reviewer have additional suggestions for empirical validation of the instruments?
>
> **Weakness 5:** We empirically evaluate the CATE estimation via PEHE in the Appendix Tables 3 and 4. These analyses are referenced in the manuscript in line 437.
>
> **Question 3:** Our network specification is highly flexible and its hyperparameters were optimized using Bayesian optimization as explained in lines 300-302. We allow for the option between non-linear, i.e. ReLU (for continuous) or temperature scaled softmax (for binary), and linear activation given the non-linear and linear semi-synthetic evaluation datasets we evaluate on. The size of the two hidden layers of each ZNet block was fixed to prevent over-fitting and reduce the parameter search space. Downstream network hidden layers can vary in size based on the Bayesian tuning.

---

> > ### Author Response · Authors · 2025-11-25
> >
> > **Question 4:** We choose the PC-based loss because in practice independence is often assessed through correlation by making an assumption of linearity. However, since this is not always true, we consider a mutual information based loss as well. We would like to highlight that the loss type is a parameter used in hyperparameter tuning as explained in lines 251-253. HSIC could be an additional option. Ultimately, we chose MI because it is a fixed rather than relative measure of non-linear variable dependence.
> >
> > We estimate the covariance and standard deviation through the obvious standard methods. There are of course potentially errors between these two statistics calculated on observed data versus their theoretical distribution in any dataset due to calculation in finite samples. However, these are consistent estimators. Could the reviewer clarify their concerns?
> >
> > **Question 5:** Continuous treatments would be conceptually the same in ZNet. Continuous treatments can be compared in bins, and our architecture automatically applies. In the case of a measurement of dose response, the downstream estimators are simply adjusted. We feel this is not conceptually interesting. Could the reviewer clarify the difference they hope to evaluate?
> >
> >
> > Whitney K Newey and James L Powell. Instrumental variable estimation of nonparametric models. Econometrica, 71(5):1565–1578, 2003.

---

> > > ### Comment · Reviewer_Y1uN · 2025-11-25
> > > **Answer to rebuttal**
> > >
> > > I thank the authors for the clarifications and references to experiments in the appendix, which partially addressed my concerns. However, major concerns still remain:
> > >
> > > - Mathematical guarantees/modeling: I do not see why ZNet necessarily learns a valid latent instrument. The loss term consists of many penalty terms, which are designed to guide ZNet to the correct direction. However, there is no guarantee that all penalties will be completely fulfilled. Therefore, there is no guarantee that the latent instrument is actually valid. The evaluation of the validity in the appendix does not show a superior performance in fulfilling the IV requirements over existing methods.
> > > - Performance: I agree with the other reviewers that the experimental results (on both ATE and CATE) do not show the superiority of ZNet. Indeed, the results across the different datasets/IVs do not show a significant difference between the methods.
> > >
> > > Overall, I do not consider ZNet a significant contribution or improvement over existing work, mainly due to the reasons stated above, and will keep my score.

---

> > > > ### Author Response · Authors · 2025-12-02
> > > >
> > > > Thank you for the attention to our clarifications! In response to your remaining concerns, we include the following additional notes:
> > > >
> > > > **Bullet 1:** We agree with the reviewer that ZNet does not necessarily learn in finite data an instrument that is perfectly valid. This lack of a finite sample guarantee results from our learning approach to identify IV representations. Here finite sample identification is impossible for any method. There is no baseline that can provide valid inference in this setting without extra knowledge of the DGP. ZNet’s training objective is a set of loss constraints, where if the constraints for exclusion restriction and unconfoundedness are perfectly satisfied, the learned IV will be valid. Given infinite data and assuming a function g exists, perfect solutions to the ZNet minimization problem give valid instruments. An autoencoder approach does not provide the same guarantee since it relies on first learning the correct probability distributions of the variables.
> > > >
> > > > **Bullet 2:** ZNet performs better in two important settings: 1) when there is influence of the unobserved confounders U on the observed data X (U \to X confounding) and 2) when there is no instrument candidate (see two Tables below). The former setting of U \to X confounding requires our additional loss component to remove instrument confoundedness. The latter setting is the most common setting since many real-world scenarios will have confounding but not a true IV candidate. We also clarify that while no single method dominates every dataset, which is unsurprising given the diversity of evaluation settings, ZNet is the strongest method on average overall. As our evaluations span heterogeneous settings, classical significance tests across these scenarios are inappropriate and the suitable metric is the average performance, which shows ZNet’s superiority for ATE estimation.
> > > >
> > > > **U->X Confounding (8 datasets)**
> > > >
> > > > | Method | DeepIV (±SE) | DF IV (±SE) |
> > > > |--------|----------------|----------------|
> > > > | ZNet | 0.198 (0.04)   | 0.205 (0.07)   |
> > > > | AutoIV   | 0.287 (0.09)   | 0.727 (0.21)   |
> > > > | VIV    | 0.317 (0.08)   | 0.399 (0.09)   |
> > > > | GIV    | 0.31 (0.07)    | 0.441 (0.14)   |
> > > >
> > > > **No Candidate (6 datasets)**
> > > > | Method | DeepIV (±SE) | DF IV (±SE) |
> > > > |--------|----------------|----------------|
> > > > | ZNet | 0.116 (0.04)   | 0.200 (0.04)   |
> > > > | AutoIV   | 0.205 (0.11)   | 0.321 (0.07)   |
> > > > | VIV    | 0.164 (0.06)   | 0.289 (0.09)   |
> > > > | GIV    | 0.256 (0.09)   | 0.402 (0.11)   |

---

### Author Response · Authors · 2025-11-25
**Overall Summary (Part I of II)**

We thank the reviewers for the detailed feedback of our paper. We appreciate the time and effort that went into these reviews, which help clarify areas for improved presentation and clarity. We take the opportunity to summarize common themes present in the reviews and list clarifications added to our revision.


### **Key clarifications and additions based on reviewer feedback:**


**Emphasis of contributions:** Reviewer 81B1 suggested we emphasize the novelty of our contribution. We highlight the following key innovations and *add language to Section 7 to emphasize these points.*


1. **Discriminatory approach to IV generation:** We proposed a novel instrumental variable (IV) generation methodology which is lightweight and high performing. In contrast to existing approaches, which learn probability distributions, we learn structural causal models through an interpretable constraint-based learning. In response to reviewers D82i and 81B1, we highlight that averaged across the comprehensive data settings, ZNet is the highest performing of the IV generation methods on ATE estimation (see table below). This is critical since in practice we cannot know the true data generation process. The table below summarizes the average across the main text Table 1 results.  *We add a figure to the manuscript appendix in revision that shows the average across all datasets for both ATE error and PEHE.*

2. **We do not assume unconfoundedness of observed data:** One critical reason that ZNet may be higher performing than existing methods is that it makes one fewer assumption, and *we add an appendix figure to clarify this distinction with causal directed acyclic graphs (DAGs).* Existing works assume that the unobserved confounders of the T, Y relationship do not influence the observed data X (i.e. assume the DAGs in the added appendix figures 8 (b), (e), (g), (i)). This allows for their losses to create unconfounded instruments by assuming the data from which they are generated is unconfounded. This assumption is untestable and unlikely. ZNet adds a loss term which eliminates confounding even in the setting that unobserved confounders influence the observed data (as in the DAGs added in appendix figures 8 (a), (d), (f), (h)). These are the settings that we highlight in the main text of the paper, and ZNet is highest performing on average across these settings.

3. **Comprehensive evaluation of IV generation:** For IV generation to be used in practice, it must perform well across untestable assumptions. In particular, it must perform well with and without the existence of a variable that can serve as an instrument in the observed data and with and without the existence of confounding across various treatment and outcome models. We conduct evaluation across all these assumptions and various SCMs. To better emphasize this contribution, *we added to the appendix DAGs illustrating the nine causal relationships, each of which corresponds to both a linear and nonlinear DGP, for a total of 18 datasets.* This is novel and critical for practical utility of IV generation methods.

| Method | TSLS      | DeepIV    | DF IV     |
|--------|---------------|---------------|---------------|
| ZNet   | **0.507 (0.252)** | **0.163 (0.041)** | **0.212 (0.059)** |
| AutoIV | 4.863 (2.469) | 0.238 (0.080) | 0.637 (0.177) |
| VIV    | 0.654 (0.183) | 0.274 (0.067) | 0.351 (0.079) |
| GIV    | 1.024 (0.353) | 0.263 (0.064) | 0.494 (0.117) |

*Rebuttal Table.* The mean absolute value of the error on ATE across datasets in Table 1 (with the standard errors across these 10 datasets) by downstream estimator and IV generation method. We remark that these data include influence of the unobserved confounders on the observed data. ZNet is the only method which theoretically constructs unconfounded instruments in this case.

*Continued below.*

---

> ### Author Response · Authors · 2025-11-25
> **Overall Summary (Part II)**
>
> *Continued.*
>
> **Clarification of assumptions and results:** It appears that some of our assumptions were misunderstood by reviewer Y1uN, leading to requests for analyses that have already been performed. We appreciate this feedback and changed wording to better present our assumptions and highlight analyses. We take the opportunity here to emphasize the following.
> - **Instruments are not assumed to exist as subsets of the observed data by ZNet.** Instead, the core contribution of our work is the generation of a representation that serves a valid instrument.
> - **We do not assume existing instruments to be normally distributed** but instead use a loss constraint to encourage normality to regulate learned representations.
> - **We evaluate the suitability of learned instruments.** Our paper evaluates the validity and strength of instruments using the standard measures of F-statistics and correlations in Appendix Tables 7, 8. *To draw further attention to these analyses, we create an additional figure for the main text illustrating a sample of these results.*
> - **We evaluate the performance of learned instruments in CATE estimation.** In addition to the ATE error results in the main text, our paper also evaluates CATE estimation in Appendix Tables 3 and 4.
>
>
> **Real world datasets:** Reviewers D82i and 3W6s suggested the addition of evaluation on “real data.” We would like to clarify that we do use real covariates in our experiments, and only simulate the  counterfactual outcomes which are never observed in any real datasets. These “semi-synthetic” datasets are a standard in evaluating causal inference methods because in semi-synthetic data, we can evaluate the true recovery of causal effects while evaluating realistic data. These data are “real” in the sense that they are generated from real world covariates. Literature in causal inference evaluates in this manner, i.e. on real-world data for which the treatment effects and outcomes are manipulated in order to be measurable. We do the same. Our datasets use the causal inference benchmark IHDP (Hill 2011) which is standard in the literature.
>
>
> **Clarification of theoretical guarantees:** Reviewer Y1uN commented for clarification on the theoretical guarantees of the validity of ZNet. We provide theoretical justification, where necessary (i.e. Lemma 1), that the ZNet loss terms enforce the properties of an instrumental variable. Solutions to the ZNet loss minimization problem will always give a function g with an output representation that serves as an instrument since IV constraints are explicitly embedded in the loss function. This instrument can then be used and evaluated like any other instrument with instrumental variable regression. Satisfying the ZNet criteria implies the validity of subsequent causal inference due to existing instrumental variable regression theory (Hartford et al 2017, Newey and Powell 2003). Beyond this, theoretical guarantees on the validity of the ZNet method for generating the best instrument amongst all suitable instruments would be asymptotic results which do not suggest practical considerations for learning. Thus, we focus on the practical and empirical evaluation of generated instruments across a variety of controlled settings. *To clarify the extent of technical guarantees, we add additional discussion in Section 7.*
>
>
>
> Whitney K Newey and James L Powell. Instrumental variable estimation of nonparametric models.
> Econometrica, 71(5):1565–1578, 2003.
>
>
> Jason Hartford, Greg Lewis, Kevin Leyton-Brown, and Matt Taddy. Deep iv: A flexible approach
> for counterfactual prediction. In International Conference on Machine Learning, pp. 1414–1423.
> PMLR, 2017
>
>
> Marno Verbeek. A Guide to Modern Econometrics. John Wiley Sons Ltd, 2004.

---

### Author Response · Authors · 2025-12-02
**Summary for ACs**

Dear ACs,

We thank you for your service providing extra time reviewing our paper and its discussion in light of recent circumstances. We propose ZNet, a novel discriminatory approach to instrumental variable (IV) generation that makes fewer assumptions and outperforms existing methods. Our evaluation was noted by reviewers to be extensive and comprehensive. During the rebuttal period we added language to emphasize our contributions and clarify the high empirical performance of ZNet in response to reviewer feedback. We also clarified several misunderstandings on our assumptions, methodology, and the nature of evaluation in causal inference on semi-synthetic data.

We take the opportunity to summarize the main remaining critique of our paper from the reviewers after our initial rebuttal.

- Reviewer Y1uN was concerned by the lack of theoretical justification. We agree that ZNet does not necessarily learn in finite data an instrument that is perfectly valid. We take a learning approach to identify IV representations where finite sample identification is impossible for any method. There is no baseline that can provide valid inference in this setting without extra knowledge of the DGP. ZNet’s loss constraints, if perfectly satisfied, learn IVs that will be valid. Given infinite data and assuming a function g exists, perfect solutions to the ZNet minimization problem give valid instruments. Beyond this, asymptotic learning bounds do not suggest practical methodology and are not included in our paper.
- Reviewer 81B1 was concerned by two of our definitions. We clarify and cite our notation in response.

We hope this brief summary helps contextualize the more detailed responses that follow. We are grateful for the reviewers’ efforts and believe the contributions of our paper remain solidly supported after resolving these misunderstandings.

Thank you,

The Authors

---

### Meta-Review · Area_Chair_Tb9L · 2026-01-06

**Summary:**

While multiple reviewers appreciate its conceptual simplicity and comprehensive empirical evaluation, there are multiple critical concerns that are not resolved after rebuttal.

1. (Y1uN, 81B1) Lack of theoretical guarantee, mathematical soundness.
2. (Y1uN, D82i, 81B1) Mixed empirical performance.
3. (D82i, 3W6s) Use of semi-synthetic datasets.
4. (81B1, 3W6s) Lack of novelty

**Reviewer Concerns:**

Concern 3. The authors explained that the use of semi-synthetic datasets is a standard practice in evaluating causal inference methods.

The authors provided more explanation for concern 1 and 2 but did not convince the reviewers.

Re concern 4, the authors provide a table to show that the proposed method requires fewer assumptions than prior works. Nonetheless this confirms the relatively limited novelty compare to prior works.

**Reviewer Scores:**

Y1uN, initial rating 2. Unlikely to change after rebutal.
D82i, initial rating 6, confidence 1. Concern 3 is resolved, and the reviewer may raise to a higher score, but the review has a low confidence level.
81B1, initial rating 2. Unlikely to change after rebutal.
3W6s, initial rating 4. Unlikely to change after rebutal.

---

### Decision · Program_Chairs · 2026-01-26

Reject